# Robust structural superlubricity under gigapascal pressures

Taotao Sun [1,2,3,7], Enlai Gao [4,7], Xiangzheng Jia [4,7], Jinbo Bian[1,7], Zhou Wang[1], Ming Ma [1], Quanshui Zheng [1,5,6] ✉ & Zhiping Xu [1] ✉

Structural superlubricity (SSL) is a state of contact with no wear and ultralow friction. SSL has been characterized at contact with van der Waals (vdW) layered materials, while its stability under extreme loading conditions has not been assessed. By designing both self-mated and non-self-mated vdW contacts with materials chosen for their high strengths, we report outstanding robustness of SSL under very high pressures in experiments. The incommensurate self-mated vdW contact between graphite interfaces can maintain the state of SSL under a pressure no lower than 9.45 GPa, and the non-self-mated vdW contact between a tungsten tip and graphite substrate remains stable up to 3.74 GPa. Beyond this critical pressure, wear is activated, signaling the breakdown of vdW contacts and SSL. This unexpectedly strong pressure-resistance and wear-free feature of SSL breaks down the picture of progressive wear. Atomistic simulations show that lattice destruction at the vdW contact by pressure-assisted bonding triggers wear through shear-induced tearing of the single-atomic layers. The correlation between the breakdown pressure and material properties shows that the bulk modulus and the first ionization energy are the most relevant factors, indicating the combined structural and electronic effects. Impressively, the breakdown pressures defined by the SSL interface could even exceed the strength of materials in contact, demonstrating the robustness of SSL. These findings offer a fundamental understanding of wear at the vdW contacts and guide the design of SSL-enabled applications.

Structural superlubricity (SSL), a state of sliding contact with nearly zero friction and essentially no wear between two solids, offers ground-breaking techniques in energy-saving and long-life tribological applications[1–11]. SSL was theoretically predicted in the 1990s by Hirano and Sokoloff[5,6,12] and later termed 'structural superlubricity' by Müser[9]. The notion characterizes the phenomenon of superlubricity induced by a structural mismatch at the atomic level. An early demonstration of SSL was made by Dienwiebel et al.[4], who measured ultralow friction at a nanoscale incommensurate contact between graphite. Experimental realization of van der Waals (vdW) SSL has been recently extended from nanoscale to microscale[2,13,14] and macroscale[3,15] sample sizes, high speeds[16], high contact pressures[14], at layered hetero-junctions[17–19], and

[1]Center for Nano and Micro Mechanics, Applied Mechanics Laboratory, Department of Engineering Mechanics, Tsinghua University, Beijing, China. [2]Railway Engineering Research Institute, China Academy of Railway Sciences Corporation Limited, Beijing, China. [3]State Key Laboratory for Track System of High-Speed Railway, China Academy of Railway Sciences Corporation Limited, Beijing, China. [4]Department of Engineering Mechanics, School of Civil Engineering, Wuhan University, Wuhan, Hubei, China. [5]Center of Double Helix, Institute of Materials Research, Shenzhen International Graduate School, Tsinghua University, Shenzhen, China. [6]Institute of Superlubricity Technology, Research Institute of Tsinghua University in Shenzhen, Shenzhen, China. [7]These authors contributed equally: Taotao Sun, Enlai Gao, Xiangzheng Jia, Jinbo Bian. ✉e-mail: zhengqs@tsinghua.edu.cn; xuzp@tsinghua.edu.cn

in a multiple-contact setup[3]. A recent breakthrough in the continuous epitaxy of single-crystal graphite films implies the opportunity to achieve SSL at length scales beyond a few centimeters[20]. These achievements shed light on device- or structure-level applications of SSL instead of a typical tip-sample setup in tribological studies.

Robustness under extreme mechanical loading conditions and long service life are crucial for practical applications of SSL, to assure reliability and endow mechanosensitive functions[21]. However, high pressures at the contact may lead to structural instabilities and result in wear. Mass loss and transfer at the sliding interface during wear may break down the SSL state and shorten the service life of the mechanical parts[22]. In 1953, Archard[23] proposed the progressive wear model, suggesting that the volume of materials removed by wear at macroscopic rough contacts is proportional to both the applied load and the sliding distance. Wear is unavoidable in this picture, although it can be minute at the beginning of the sliding process. Recently, the atom-by-atom attrition model for microscopic wear was developed[24,25], indicating that the theory of progressive wear applies to both macroscale[26] and microscale systems[27]. It is worth noting that recent experimental evidence demonstrates a wear-free behavior over the 100 km sliding distance at an SSL contact[28], indicating that damage activation and accumulation may be absent. The picture of progressive wear thus might fail at the SSL state. The pressure on the contact in these studies is on the order of several megapascals, much lower than the extreme conditions in applications (e.g., the strength of materials), which could reach the gigapascal level. The vdW interface such as that between graphite remains stable even under the pressure of several tens of gigapascal before structural transitions into diamonds[29–34]. It would thus be interesting to probe the upper bound on the admissible pressure of an SSL state and explore the potential failure mechanisms at the interface beyond the bound, where the progressive wear mechanism may be recovered.

We explored the wear characteristics of SSL by studying two representative vdW contacts of graphite/graphite ('self-mated') and tungsten/graphite ('non-self-mated') at elevated contact pressure using a home-built loading system[35]. The critical pressure ($P_{cr}$) below which wear is absent was characterized experimentally, which could reach the gigapascal level and is shown to be strongly tied to the nature of interfacial electronic coupling. The microscopic process of wear identified in experiments above $P_{cr}$ was analyzed by atomistic simulations, suggesting a step-wear mechanism that can activate subsequent progressive wear processes. The study demonstrates the outstanding mechanical robustness and wear-free feature of SSL under pressure even beyond the strength of the materials in contact and lays down the principles of SSL design in tribological or device applications via material selection and pressure control.

## Results

### SSL at graphite contacts under high pressure
Wear characteristics of SSL were first explored at the graphite/graphite contact because graphite is the most commonly used material for SSL (Fig. 1). Figure 1a, b illustrates the experimental setup and an optical microscopy (OM) image of the contact constructed by cleaving and transferring a self-retractably moving (SRM) flake from a microscopic graphite mesa (the '*mesa*')[13] onto a mechanically exfoliated graphite flake (the '*substrate*'). Our previous works showed that any cleaved SRM flake from highly oriented pyrolytic graphite (HOPG) features single-crystalline surfaces without detectable defects, as characterized by atomic force microscopy (AFM), electron backscattered diffraction (EBSD), and Raman spectrum techniques[19,36,37]. Crystal orientations of the graphite substrate were studied using EBSD (Fig. 1g–l), and the surface roughness was characterized by OM and AFM before tests. The results confirm that the surfaces in contact are single-crystalline and free of grain boundaries. During the tests, normal loads were applied to the mesa through a tungsten tip with a radius of several

micrometers. A home-built loading system is used to apply higher pressure over a larger contact area than the nanoscale contacts studied using the AFM-loading system[35]. The loading amplitude was controlled in a closed loop. The graphite substrate was driven through a displacement stage, which leads to relative sliding at the contact. All experiments were carried out under the ambient environment and were in-situ monitored through an OM. The friction stress between the mesa and the substrate is ultralow (~20 kPa), and the friction coefficient (the ratio of the friction force and the normal force) is on the order of $10^{-5}$ under pressures between $P = 1–6$ GPa (Fig. 1c). The nearly pressure-independence of friction forces measured up to the gigapascal scale demonstrates the robustness of SSL at the graphite/graphite contact under high pressures.

Figure 1d, e shows the OM and AFM images of the graphite substrate after a sliding test under a normal load of 10 mN (the highest accessible normal load by the loading system which corresponds to a contact pressure of $P = 9.45$ GPa (see Supplementary Note 1 for details)) and a sliding distance up to 10 mm in $5 \times 10^2$ cycles. The size of effective contact (~0.8 μm) between the graphite mesa under the tip and the substrate is much smaller than the size of the mesa (6 μm × 6 μm). The edge effect[38] of the mesa on the robustness of SSL can be neglected. By further considering the atomic-level flatness of graphite, the roughness effect on the nature of local contact is expected to be minor. A pre-cleaning step was carried out to sweep out contaminants (e.g., adsorbed molecules such as water and hydrocarbons[39–41], see Supplementary Note 2 for details). The contaminants originate from the environment before the construction of the contact and may not be completely excluded at the contact. However, the ultra-low friction coefficient of SSL is still preserved, indicating the robustness of SSL against the atmosphere[41]. Consequently, the pre-cleaning step before tests removes some of the confined molecules, reducing friction to a steady-state level (Supplementary Fig. 3b). Our previous work reported no obvious difference in friction of SSL in the mesa/substrate setup in ambient conditions and nitrogen atmosphere with relative humidities (RHs) of 42% and 10%, respectively[19]. After the test, debris of contaminants was characterized at the boundaries of the pre-cleaned region as a result of the edge-sweeping process. In contrast, neither aggregation of debris nor rupture of the material was observed in the region under sliding tests, suggesting that the contact may be contaminant-free. The resolution of our AFM characterization is 0.1 μm × 0.1 μm × 1 nm (Fig. 1e), which concludes the absence of wear below a minimum detectable wear rate of $10^{-10}$ mm³/Nm under a normal load of 10 mN and the sliding distance of 10 mm. In comparison, the wear rates at a macroscopic steel contact and the microscopic silicon/silicon nitride contact are $10^{-7}–10^{-3}$ mm³/Nm[42] and $10^{-6}–10^{-4}$ mm³/Nm[43], respectively. Raman spectroscopy characterization also confirms the absence of the D peak at 1350 cm⁻¹ in the graphite substrate (Fig. 1f), suggesting no detectable atomic-level defects. These results demonstrate the exceptional wear resistance of self-mated graphite SSL under gigapascal-level pressures.

### Breakdown of SSL at non-self-mated vdW contacts
The breakdown of graphite/graphite contact under high pressures proceeds with interfacial failure of the material itself. Notably, our recent efforts made it possible to construct SSL contacts with only one of the contact surfaces from the vdW-layered materials. Theoretical calculations demonstrate that the non-self-mated SSL contacts between metal and graphite exhibit weaker pressure resistance than the graphite/graphite contacts[44], and it is natural to question the robustness of the SSL state therein. A tungsten/graphite contact was designed as a testbed to explore the breakdown and wear processes at a non-self-mated SSL contact. Tungsten is chosen for its high strength among common metals widely used in mechanical systems. The contact was constructed by pushing a tungsten tip with a radius of several micrometers onto the graphite substrate (Fig. 2a, b). Normal forces

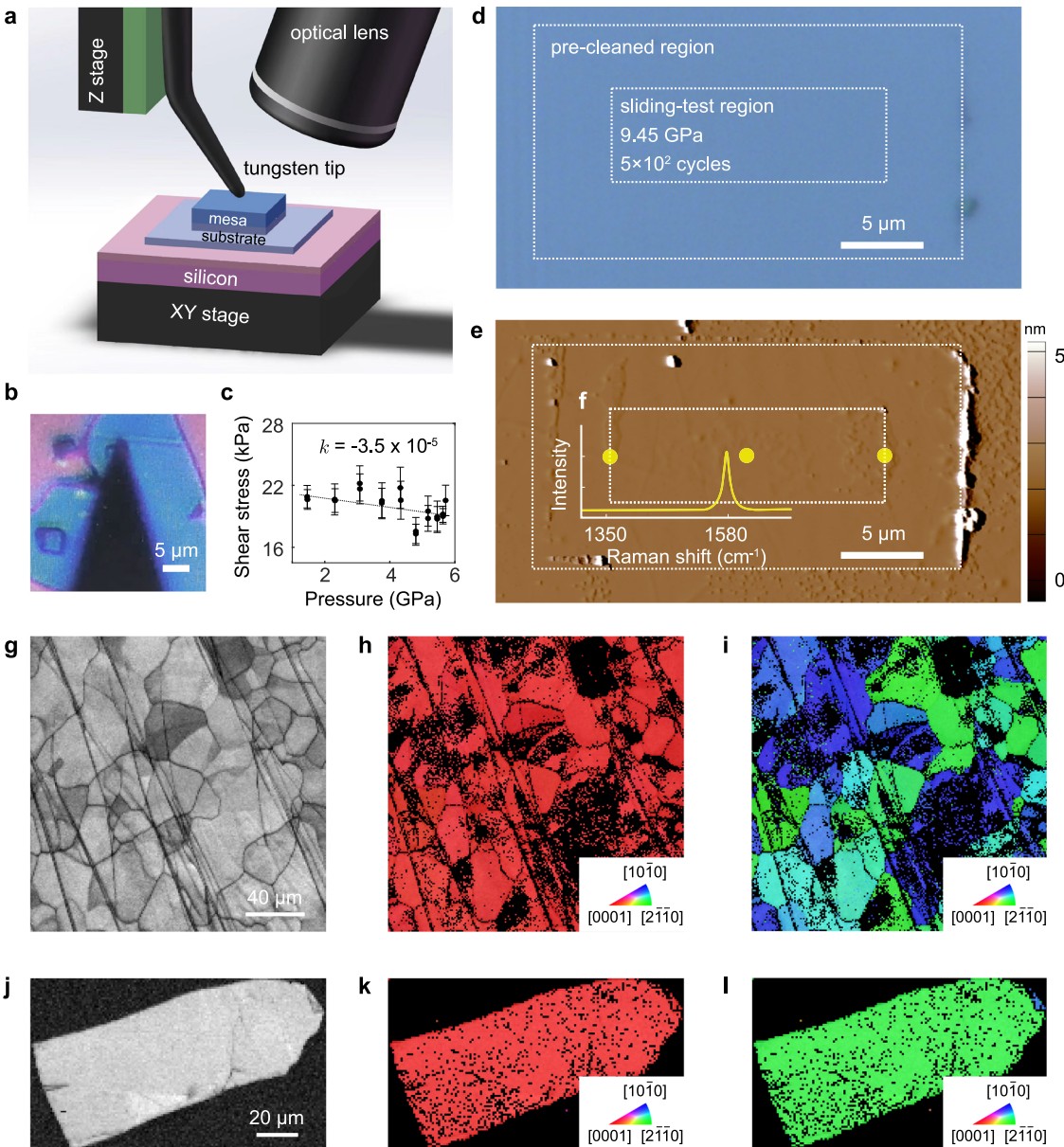

**Fig. 1 | Wear tests of the graphite/graphite contact. a, b** Experimental setup (**a**) and optical microscopy (OM) image (**b**) of the graphite/graphite contact in the mesa/substrate setup. **c** Experimental measurement of the average shear stress below the breakdown pressure. The results include both loading and unloading processes. The error bar represents the standard deviation of 10 repeated experiments, and $k$ is the fitting slope. **d, e** OM (**d**) and atomic force microscopy (AFM) images of the graphite substrate after sliding tests. The testing procedure includes 2 steps of (1) pre-cleaning the substrate within a region of 26 μm × 14 μm (the large dashed box), and (2) conducting sliding tests at the center of the cleaned region under the pressure of 9.45 GPa with a reciprocating sliding amplitude of 10 μm and a sliding velocity of 10 μm/s (the small dashed box). The sliding distance reached 10 mm in $5 \times 10^2$ cycles. **f** Raman characterization of the graphite substrate at the marked points. **g–i** Band contrast (BC, **g**) and inverse pole figures (IPFs, **h**, **i**) of HOPG. **j–l** BC (**j**) and IPFs (**k, l**) of normal flake graphite. Source data are provided as a Source Data file.

were applied to the tungsten tip through the home-built loading system. Other experimental setups follow the graphite/graphite contact. The friction coefficient is measured to be on the order of $10^{-3}$ even under gigapascal-level pressures (Supplementary Fig. 5). However, the shear strength, $\tau_s = 10$ MPa, is much higher than that of the graphite contact (~20 kPa) possibly due to the adhesion effect. Consequently, the notion of SSL should be justified if the mechanical energy dissipation during the sliding process and thus the shear strength are of concern.

Wear tests of the tungsten/graphite contact were conducted in an increasing load sequence from 0.1 mN to 10 mN (see "Methods" section for details). Figure 2c, d shows the OM and AFM images of the graphite substrate after sliding tests under loads of 0.2 mN and

0.5 mN, which corresponds to a pressure of $P = 3.74$ GPa and 5.07 GPa, respectively (Supplementary Fig. 1). The sliding velocity is 10 μm/s with a reciprocating amplitude of 30 μm. Indication of wear is absent after a sliding distance of 0.6 m, or $10^4$ cycles, under $P = 3.74$ GPa, but emerges as $P$ increases to 5.07 GPa. The long-distance wear-free performance of SSL contacts under high pressures cannot be explained by the progressvie wear model which predicts that the mass loss is proportional to the pressure and sliding distance. This is because the SSL contact is atomically smooth and the in-plane $sp^2$ bonding network is strong[45,46]. The result suggests that a critical pressure ($P_{cr} > 3.74$ GPa) is required to activate the wear process (see Supplementary Note 4 for more experimental results). AFM characterization of the test region shows that wear of the tungsten/graphite contact occurred at the beginning

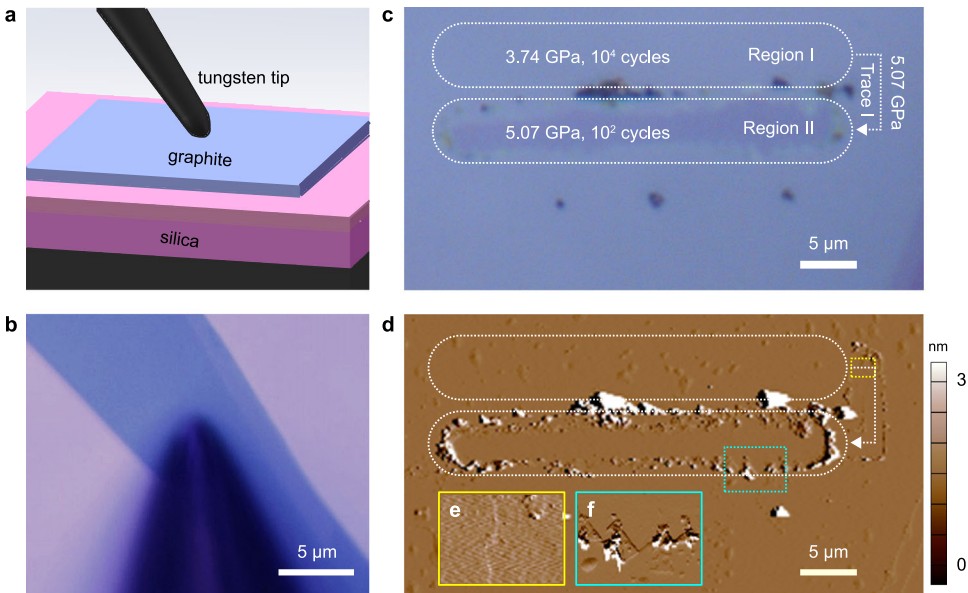

**Fig. 2 | Wear tests of the tungsten/graphite contact. a, b** Experimental setup (**a**) and OM image (**b**) of the tungsten/graphite contact. **c, d** OM (**c**) and AFM (**d**) images of the graphite substrate after sliding tests. The testing procedure includes 3 steps of (1) sliding the tungsten tip under the pressure of 3.74 GPa within Region I, (2) elevating the pressure to 5.07 GPa and move the tip from Region I to Region II through Trace I, and (3) sliding the tip under the pressure of 5.07 GPa within Region II. The sliding velocity and sliding amplitude are 10 μm/s and 30 μm, respectively. The test was stopped once obvious wear of graphite was in-situ observed by OM, or the number of sliding cycles reached $10^4$ (corresponding to a total sliding distance of 0.6 m). **e, f** Magnified views of the step edge indicated by yellow and cyan dashed boxes in **d**. Source data are provided as a Source Data file.

of sliding under $P = 5.07$ GPa (Trace I, Fig. 2c), which proceeds by tearing of graphite layers (Fig. 2e, f).

## Mechanisms of SSL breakdown and wear activation

The breakdown pressure of the tungsten/graphite contact is higher than 3.74 GPa, beyond which wear was initially observed. To elucidate the breakdown and wear mechanisms of SSL represented by vdW contacts under pressure, we carried out atomistic simulations. The SSL contact shows a step-wear scenario where material loss is not activated below the critical pressure, in contrast to classical laws of progressive wear. Two key stages are proposed following the experimental evidence to include (I) wear initiation through pressure-assisted interfacial bonding, and (II) wear development through shear-induced tearing of the graphite layers.

It is noted that different from the conventional rough contacts with multi-asperities[45,46], the SSL contact studied in this work is atomically smooth and can be described as a single-contact model by considering the stiff in-plane bonding network. Previous work[47] showed that the tungsten tip etched by KOH is smooth, with a root mean square (RMS) roughness below 0.3 nm. We characterized our tip using high-resolution SEM and AFM. The results show height variation of a few nanometers at the micrometer scale and atomistic smoothness at the nanometer scale (Supplementary Fig. 6c, d), which validates the argument that local roughness is not a crucial issue for the discussion. A recent work[48] shows that the contact pressure change converges as the roughness wavelength increases. Therefore, the effect of roughness on contact pressure is negligible. Our indentation setup in the sliding tests results in a local high pressure across an effective contact region of hundreds of nanometers (Supplementary Note 1). Wear at the SSL contact can be directly related to the pressure enforced across the interface, the mechanochemistry of which was explored by first-principles calculations based on the density functional theory (DFT). X-ray photoelectron spectroscopy (XPS) characterization of the tungsten tip suggests the existence of an oxide layer over the tip surface (Supplementary Fig. 6f). A model consisting of a $(\sqrt{2} \times \sqrt{2})$R45°-reconstructed (001) $WO_3$ surface and a graphite

substrate was adopted, which closely follows our experimental setup and previous studies[49,50] (Fig. 3a, b). The DFT calculation results show that the pressure increases rapidly with the compression and declines after the peak at a pressure of 3.50 GPa (Fig. 3d), which is defined as the breakdown pressure ($P_{cr}$) due to the formation of interfacial O-C bonds (Fig. 3c). The value of estimated critical pressure agrees with the experimental value of 3.74 GPa.

To quantify the change in interfacial electronic coupling across $P_{cr}$, we calculated the electron localized function (ELF) that measures the extent of spatial localization of the reference electron[51]. The value of ELF ranges from 0 and 1. Perfect electron localization and free electron gas behaviors are identified by ELF values of 1 and 0.5[52], respectively. According to our model, the ELF value between the nearest O-C atom pairs is less than 0.1 for $P < 1.7$ GPa, which suggests typical vdW interaction. As the pressure reaches $P_{cr}$, this value exhibits an increase to 0.27, indicating the rise of ionic bonding characteristics at the interface[21]. Beyond $P_{cr}$, the value of ELF increases to 0.79, suggesting the formation of stable covalent bonding between O and C atoms, which competes with the strong in-plane bonding network and results in wear (Fig. 3c). The effects of pressure-assisted bonding on the frictional characteristics were then explored. Before the formation of interfacial O-C bonds, shear strengths calculated by first-principles simulations remain as low as < 0.14 GPa (Fig. 3e), indicating the ultra-low shear resistance at the $WO_3$/graphite interface below the breakdown pressure. The shear strength increases to $\tau_s = 3.65$ GPa after $P_{cr}$ is reached (Fig. 3e). Molecular dynamics (MD) simulations show that wear is activated by the formation of wrinkles and tears in the graphite layers caused by shear forces at the interface (Fig. 3f). The deformation and failure of the graphite layer near the trailing edge can be predicted by the shear-lag model, which predicts strain localization at the contact edge[53]. This finding explains our experimental observation of wear in the form of graphite tearing.

The same argument applies to the graphite/graphite SSL contact although the value of $P_{cr}$ is not experimentally determined. Previous studies show that the vdW interface between graphite remains stable even under the pressure of several tens of gigapascal in prior to

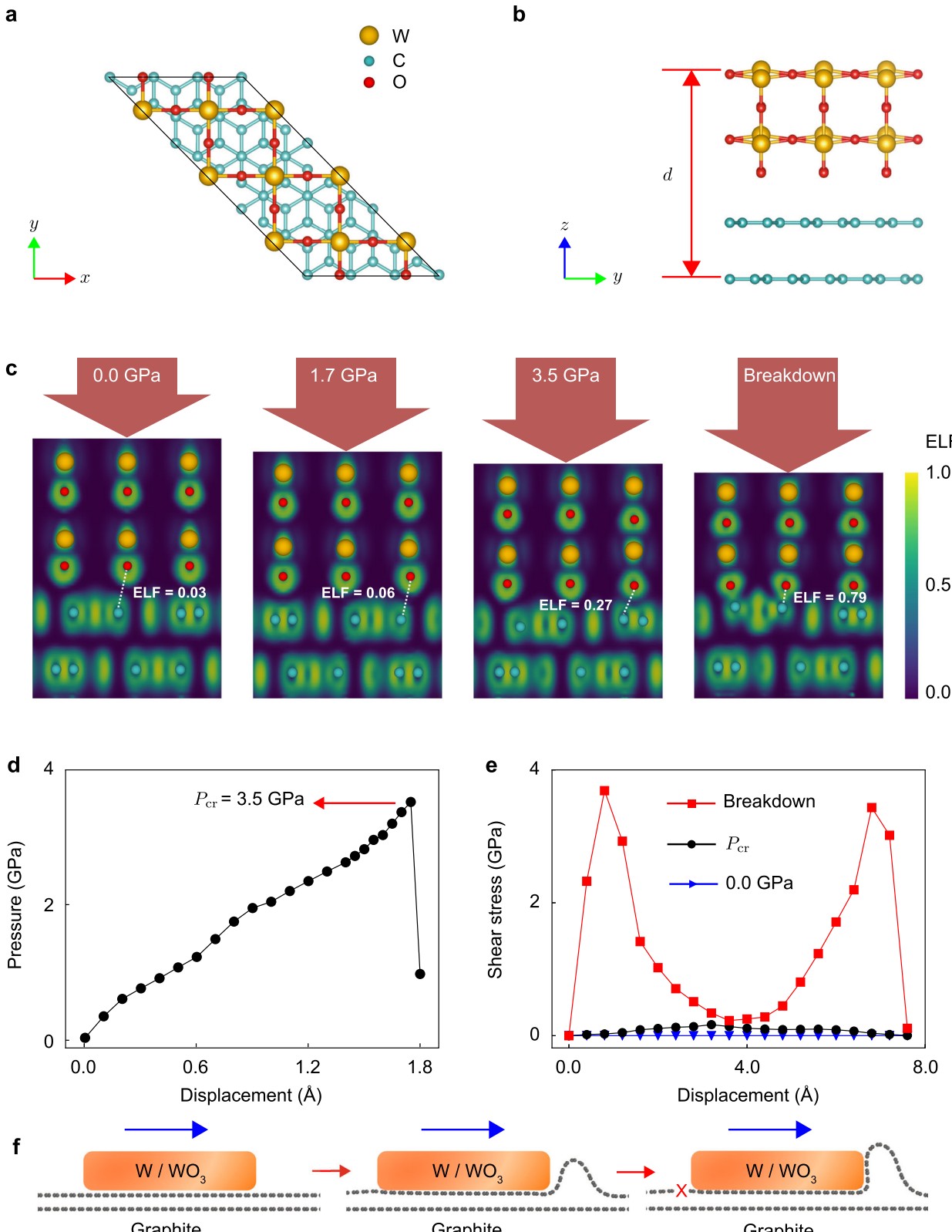

**Fig. 3 | Pressure-assisted bonding and wear at the WO₃/graphite interface.**
**a**, **b** Model of the WO₃/graphite contact. **c** Electron localization function (ELF) showing the evolution of structural responses and interfacial bonding states with pressure. The dashed lines show the atom pairs with the strongest interaction. **d** Pressure-displacement relation obtained from density functional theory (DFT) calculations. **e** Shear stress-displacement relation at different pressure levels. **f** The step-wear process demonstrated via molecular dynamics (MD) simulations, where pressure-assisted bonding triggers wear through shear-induced tearing. The puckering effect at the contact front is caused by accumulated in-plane deformation of the graphene layer, which prefers to bend instead of being compressed. Source data are provided as a Source Data file.

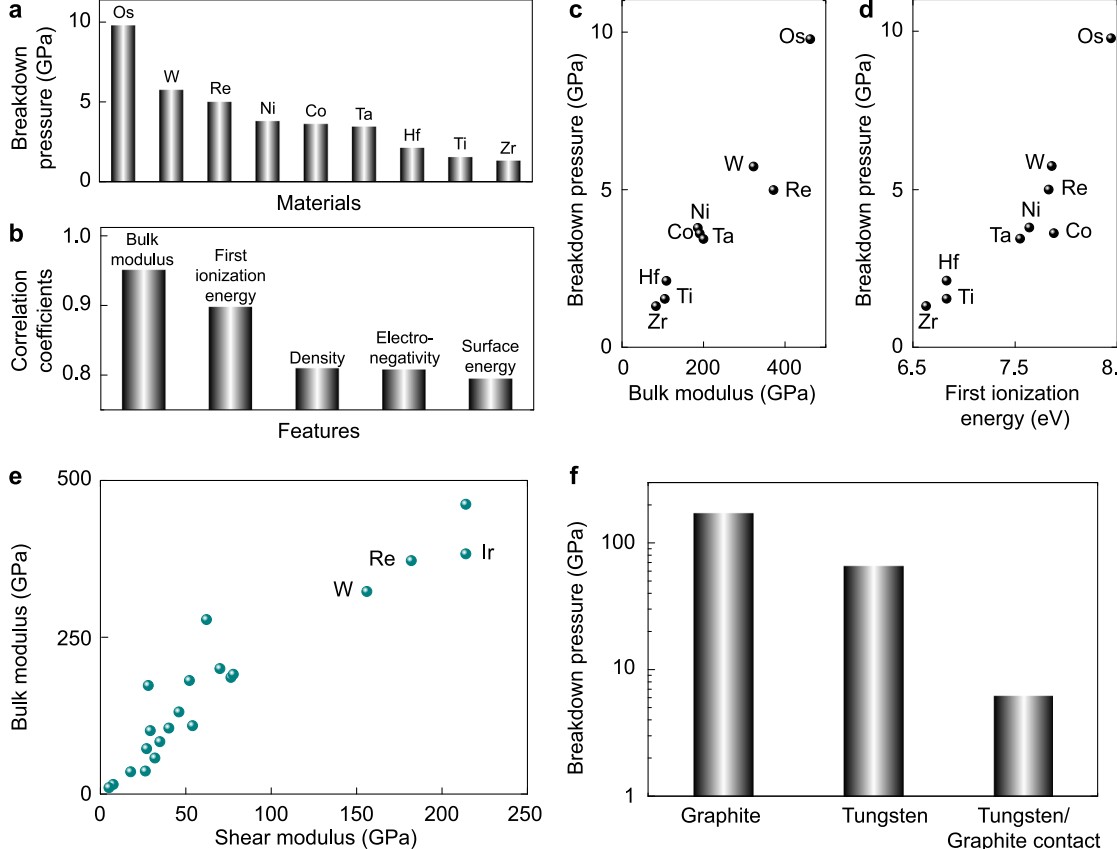

**Fig. 4 | Physics behind the breakdown pressures. a** Breakdown pressures calculated for metals Os, W, Re, Ni, Co, Ta, Hf, Ti, and Zr in contact with graphite. **b** Top-ranked features that strongly correlation with the breakdown pressure. **c**, **d** Relation between the breakdown pressure and bulk moduli (**c**), and the first ionization energy of the metal (**d**). **e** Bulk moduli and shear moduli of the metals. **f** Breakdown pressures of graphite, tungsten, and the tungsten/graphite contact. Source data are provided as a Source Data file.

structural transitions into diamonds[29–31]. The highest accessible pressure in our experimental setup is only 9.45 GPa, below which no bonds are formed across the interface and wear can hardly be activated. Although pressure loading in our work is different from the hydrostatic compression used to explore structural transitions from graphite to diamond, the underlying mechanism remains similar, that is, the transition from $sp^2$ to $sp^3$ bonding networks. The reported gigapascal-level pressures for this transition thus provide support for the ultrahigh breakdown pressure of SSL we uncovered. To verify the step-wear mechanism at the graphite/graphite contact, vacancy defects were introduced into the graphite substrate by argon plasma treatment. The experimental results show that the breakdown pressure of the defective graphite/graphite contact is reduced to 0.4 GPa and wear is characterized by tearing ruptures of the graphite substrate (Supplementary Note 6). These results suggest that the step-wear mechanism also applies to the graphite/graphite contact. This finding agrees with the fact that solid lubrication using graphite should avoid the formation of interlayer bonding in dry or vacuum conditions, where water can solve this problem by providing -H and -OH terminations when C-C bonds break.

## Understanding SSL robustness under high pressures

Our work demonstrates a wear-free feature of the graphitic SSL contacts without interfacial bonds under GPa pressures over long sliding periods. To understand the unexpected robustness of SSL states and extend our discussion to SSL-enable applications, we studied material dependence. A W/graphite contact can be constructed by preventing oxidation and was studied by performing DFT calculations for comparison with the WO$_3$/graphite and graphite/graphite contacts (see "Methods" section for details). The DFT calculation results suggest a

higher breakdown pressure of $P_{cr} = 5.4$ GPa for the contact with bare W (001) surface. We find that instead of covalent bonding at the WO$_3$/graphite contact beyond $P_{cr}$, the transition in the electronic coupling at the W/graphite interface is mediated by charge transfer[54,55], and this electrostatic nature of interaction results in a higher value of $P_{cr}$.

In contrast to the WO$_3$/graphite contact, the W/graphite interface is electrically conducting and thus more interesting for device applications (Supplementary Fig. 6e). Our discussion is elaborated by including a wide spectrum of metals in contact with graphite (Fig. 4). DFT calculations report the breakdown pressures and identify two characteristic modes of failure beyond it. The first class of non-self-mated contact (e.g., Cu, Au, Ag/graphite) can withstand pressure up to ~100 GPa, which is much higher than the compressive strength of the metals themselves where plasticity is triggered[53,56]. As a result, the breakdown pressure of the SSL contact is limited by the strength of metals instead. For graphite contacts with Os, W, Re, Ni, Co, Ta, Hf, Ti, and Zr, the vdW interfaces break down by forming covalent bonds. The value of $P_{cr}$ is much lower, with the highest value of 9.78 GPa for the Os/graphite contact (Fig. 4a). The underlying physics behind the pressure resistance can be elucidated by the following understanding of cohesion in solids. It is well known that the 'physical' stiffness of a solid is strongly tied to the 'chemical' one defined by the ionization energy (IE) and the electron affinity (EA)[57]. This understanding is extended to the vdW interface here. A correlation analysis shows that the most relevant materials features to the value of $P_{cr}$ are the bulk modulus ($B$) and IE, where the coefficients of correlation are 0.95 and 0.90, respectively (Fig. 4b–d). Metals with high elastic moduli usually feature high surface electron densities, which lead to higher resistance to the transition in the electronic coupling. On the other hand, metals with higher IEs are less reactive, and thus higher

pressures are needed to form chemical bonds between the metals and graphite. The destruction of SSL contact under pressure is thus a result of the combined effects of structural distortion in the metals and charge transfer at the interface if the metals are stable by themselves. This understanding can guide material screening robust SSL applications under high pressure.

In brief, the robustness of SSL states under gigapascal-level high pressure is reported here. Below the breakdown pressure, the interfacial electronic coupling can be tuned by the pressure but signature transitions are not present. The findings are important for tribological applications under extreme loading conditions, and reconfigurable device applications by opening the avenue of pressure control. Specifically, reconfigurable SSL-enabled devices can be constructed by harnessing the sliding motion. The robustness of SSL at graphite/graphite and tungsten/graphite (WO$_3$/graphite and W/graphite) contacts are studied in our experiments under ambient conditions with a sliding velocity of 10 μm/s. The speed range covered by common friction tests conducted by the AFM and tribometers is $10^{-7}$–$10^{-2}$ m/s[3,19]. Reportedly, SSL can be sustained under the speed from 25 m/s[16] to 294 m/s[58]. Temperature is another crucial factor in practical applications. Thermal fluctuation not only facilitates sliding over free energy barriers and results in reduced friction[59,60], but also helps to activate the breakdown of SSL interfaces that can be regarded as a chemical reaction[24,25].

## Methods

### Sample preparation
The graphite mesa (6 μm × 6 μm) was etched from the highly oriented pyrolytic graphite (HOPG) using the oxygen plasma, before which a SiO$_2$ film with a thickness of 100 nm was deposited on top of the HOPG for increasing the bending stiffness and the friction resistance during the tip manipulation. The graphite substrate was mechanically exfoliated from the normal-flake-graphite (NGS, Germany) by the Scotch tape method and transferred onto a silicon substrate with a 300-nm-thick SiO$_2$ layer. The graphite/graphite contact is constructed by transferring the graphite mesa onto the graphite substrate (the mesa/substrate setup) using a tungsten tip manipulated by a micromanipulator (Kleindiek MM3A)[19,61]. Tungsten tips with a radius of a few micrometers were electrochemically etched in the KOH solution (5 mol/L) with a reaction expression as, W + 2KOH + 2H$_2$O → K$_2$WO$_4$ + 3H$_2$↑[47,62], where the reaction product K$_2$WO$_4$ is soluble (solubility of 51.5 g/100 g H$_2$O@20 °C). All tips were ultrasonically rinsed with acetone, alcohol, and deionized water sequentially before conducting tests to exclude the effects of adatoms and oxides. Both graphite surfaces at the contact are single crystalline (Fig. 1g–l).

### Wear tests
Experiments were conducted in a home-built loading system[35], where the loading range of the system is 0.1–10 mN. The amplitude of the loads can be closed-loop controlled during sliding. Forces were calibrated by a high-precision balance (METTLER TOLEDO, XA205DU) before tests. All tests were conducted in an increasing load sequence of 0.1, 0.2, 0.5, 1, 2, 3, 5, 7, 9, and 10 mN with the same sliding velocity of 10 μm/s. To determine the critical pressure, $P_{cr}$, the number of sliding cycles under each load is set to 10. For long-distance sliding tests, $5 \times 10^2$ cycles were carried out for the graphite/graphite contact to ensure a distance of sliding over 10 mm, and $10^4$ for the tungsten/graphite contact with a distance over 0.6 m. The loading system is equipped with an OM (Hirox KH-3000) to locate the tip to the microscale mesa or the substrate.

### Wear characterization
Wear was first judged by in-situ OM (Hirox KH-3000) characterization, which can monitor the change of the surfaces in real-time. Detailed wear characterization was conducted by using an AFM (Oxford Instrument MFP-3D Infinity) in the tapping mode. Morphology changes of surfaces were measured by the vibration amplitudes of the AFM probe. Raman spectroscopy characterization (HORIBA Scientific) was carried out to quantify the atomic-scale defects.

### Friction measurements
Friction at the graphite/graphite contact was investigated by using a home-built two-dimensional force sensor. The lateral resolution of the sensor is ~80 nN and the range of the normal load is on the order of milli-Newtons[35]. Normal loads were applied to the SiO$_2$ cap on top of the graphite mesa in a closed-loop control. Friction was measured at a sliding speed of 10 μm/s, which was repeated for 10 cycles.

### First-principles calculations
To obtain the breakdown pressure and shear characteristics between a tungsten tip (W or WO$_3$ if surface oxidation is considered) and graphite, DFT-based calculations were performed by using the Vienna ab initio simulation package[63]. The generalized gradient approximation (GGA) of the Perdew-Burke-Ernzerhof parameterization (PBE) was used to describe the exchange-correlation functional[64,65]. A cutoff energy of 520 eV was used for the plane-wave basis set. The vacuum layer was set as 4 nm to avoid the interaction from the periodic images[66]. For Brillouin-zone integration, the Monkhorst-Pack $k$-grid with a mesh density of 3 Å$^{-1}$ was adopted. The structures were relaxed by using the conjugated gradient (CG) algorithm. The threshold for energy and force convergence was set as 0.1 meV/atom and 0.01 eV/Å, respectively.

The computational supercell consists of 2 layers of graphene in AB stacking or 2 (4) atomic layers of W (WO$_3$). To reduce the size effect, we used supercells (3 × 1 W/2 × 4 graphite, 2 × 3 WO$_3$/3 × 5 graphite), where the lattice misfit of graphite is 2% and 3%, respectively. Periodic boundary conditions (PBCs) along the in-plane directions were enforced. The breakdown pressure was studied by moving the mesa (W or WO$_3$) towards the graphite substrate stepwisely, where the top layer of the mesa and the bottom layer of the substrate are fixed. DFT calculations were performed to determine the pressure from the forces acting on the atoms in the mesa, as a function of the interfacial distance at the contact[21]. The breakdown pressures were determined from the peaks in the pressure-displacement curves. Shear tests were performed by transversely moving one of the contact surfaces. The shear stress is calculated from the forces acting on the top layer of the mesa along the sliding direction. To calculate the compressive strengths or breakdown pressures of metals, 4 atomic layers were constructed.

### Molecular dynamics simulations
Molecular dynamics (MD) simulations were carried out using the large-scale atomic/molecular massively parallel simulator (LAMMPS)[67]. The all-atom optimized potential, which can successfully capture essential interatomic interactions, was adopted to describe the interatomic interactions for graphite[68]. The vdW interaction was described by the 12–6 Lennard-Jones potential $V(r) = 4\varepsilon[(\sigma/r)^{12} - (\sigma/r)^6]$ with a cutoff distance of 1.2 nm. At a reduced interfacial distance of 2.5 Å, the shear strength (5.22 GPa) exceeds the breakdown pressure. PBCs along the in-plane directions were used in all simulations. All constructed structures were fully energy-minimized using a conjugate-gradient algorithm before the shear test. Shear was applied by moving the tungsten layer at a velocity of 20 m/s, and the mechanical responses were investigated at 0.1 K using a Nosé-Hoover thermostat. Two edges of the graphene layer were fixed to avoid rigid displacement of the graphite.

## Data availability
All data generated in this study are provided in the Source Data file. Source data are provided with this paper.

## Code availability
All codes used in this study are available from the corresponding author (Z.X.) upon request.

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

## Acknowledgements
This work was supported by the National Natural Science Foundation of China (12425201, 52090032, 11890672, 12172261, 11832010, and 11921002). The numerical computation was performed on the Explorer 1000 cluster system of the Tsinghua National Laboratory for Information Science and Technology, and the supercomputing system in the Supercomputing Center of Wuhan University. T.S. wishes to acknowledge Mr. Maosheng Chai for his help on AFM and Raman characterization and Dr. Jin Wang for fruitful discussion.

## Author contributions
Q.Z. and Z.X. conceived and supervised the research. T.S. performed the experiments. E.G., X.J., J.B., and Z.W. performed the simulations and theoretical analysis. T.S., E.G., X.J., J.B., M.M., Q.Z., and Z.X. participated in the discussion. T.S., E.G., J.B., and Z.X. wrote the manuscript.

## Competing interests
The authors declare no competing interests.
