## [Peer Review File · Nature Communications]

REVIEWER COMMENTS

Reviewer #1 (Remarks to the Author):

In this manuscript, the authors present a hybrid study that combines both experimental and theoretical approaches to explore the outstanding robustness of SSL under extremely high pressures up to the pressure limit. The paper thoroughly investigates both the symmetric vdW contact between graphite with incommensurate interfaces and the asymmetric vdW contact between a tungsten tip and graphite substrate. These results provide valuable insight for design SSL with enhanced pressure-resistance. Overall, this research is quite interesting and useful for the readers from the tribology community. There are some issues that require further clarification and explanations before publishing the paper.

1. The atomically smooth contacts of SSL studied in this work markedly differ from traditional rough contacts. This contrast might account for why the pressure-resistance of common rough contacts falls short of the SSL contacts. Please discuss the difference of pressure-resistance in these contacts.
2. The authors investigated the correlation of the breakdown pressure with bulk modulus. This is a quite interesting, since a bulk property is linked with an interfacial property. However, more explanations should be provided for better understanding this behavior.
3. In many contacts, the materials involved are nearly isotropic, whereas the graphite used in SSL contacts is significantly anisotropic. How does this anisotropy affect the pressure-resistance and its wear behaviors?
4. The reported breakdown pressure of SSL, reaching up to several GPa, exceeds the actual compressive strength of many materials with defects. Accordingly, a discussion regarding the influence of material defects on the pressure resistance of SSL contacts would be beneficial.
5. If the pressure exceeds the breakdown pressure and is subsequently removed, is it possible for the covalent interface recovered to vdW contacts?
6. More details on the experimental determination of the breakdown pressure should be provided.

Reviewer #2 (Remarks to the Author):

The paper describes high pressure experiments of layered materials. The authors find that superlubricity conditions can be maintained up to 9GPa in the case of graphite mesa on graphite and 3GPa for tungsten tip on graphite. The analysis of a variety of metals on graphite points to the influence of the bulk modulus and the ionization energy to be the most important parameters to have a high threshold for the transition to wear conditions. Below the threshold the authors report wearless friction. This is a quite important observation, which differs from the more common wear models, which assume a continuous removal of atoms from the wear zone. Therefore, superlubricity conditions seem to be suitable to drastically reduced wear at relatively high pressures. In summary, I support publication of this important observation.

Minor points:

- It would be interesting to compare the observed absence of wear with typical wear rates under similar conditions of more common materials.
- It might be also of interest to explore the temperature stability of superlubricity conditions. Some comments on this point might be of interest to the reader.
- It would be also interesting to comment on the influence of humidity.

Reviewer #3 (Remarks to the Author):

Sun et al. present a study combining experiments and atomistic modeling, in which they find low friction in two systems to remain robust up to Gigapascal stresses. Since others have found robust superlubricity in similar systems, e.g., Ref. 17, 20, 21, and 22, I find the reported progress somewhat incremental. In fact, Web Of Science identifies 18 papers with the keyword "robust superlubricity" in the title and most of them are concerned with graphite and/or graphene-terminated surfaces under a relatively large compressive stress, just like the submitted manuscript. Thus, the robustness of superlubricity is no news.

The potential novelty of this paper is the attempt to rationalize the breakdown of graphite's superlubricity. However, I find the connection between the DFT calculations and the experiments rather weak. What the authors appear to find is a hybridization instability of carbon atoms from sp² to sp³ when being squished against a metal, which could be perceived as an adhesive buckling instability. However, the authors failed to investigate if this instability is reversible when the normal stress is removed. If they did observe mass transfer under decompression, they could have made a stronger case that the simulations pertain to the experiments. This way, the authors only find an instability, which, however may very well be elastic in nature.

What I found truly disappointing is that this hybridization induced buckling instability, which is a purely quantum mechanical effect, was related to the well-known puckering effect, which is related to the classical, thin-sheet mechanics of a graphite layer. I see little relation between the two. The one aspect that does make sense, however, is that the onset of buckling is related to the question how readily graphite's counter face is donates electrons to graphite.

The part that I found the most disappointing, however, is that the friction reported from DFT is computed only over a sliding distance of 1 Angstrom. To get reliable numbers, shear forces must be averaged over periodic patterns, e.g., multiple integers of the lattice constant, or, in disordered or symmetry broken systems, over a meaningfully long sliding distance. The friction force reported in Fig. 3d is about to become negative so that it is absolutely impossible to even crudely estimate a mean frictional stress from that graph. It would be very interesting to see how the sp³ hybridization state of carbon atoms does get dragged along and if/how it progresses to the next atom. If there were instabilities, then the breakdown of superlubricity could be argued to arise from the sp² to sp³ hybridization changes, but the relation to wear would nonetheless still be missing.

To make the case for wear, one must study the contact edges, in particular the stress distribution near the trailing edge, where tensile stresses are largest. Such an analysis was not presented in the given work and requires advanced modeling skills.

In conclusion, I do not see the manuscript to meet the criteria of Nature Communications. The experimental results are sufficiently interesting to be disseminated in a good, regular journal, while the analysis of the simulation results (which certainly are interesting) would require a much cleaner analysis of the observed instability and its effect on friction and wear.

Reviewer #4 (Remarks to the Author):

The report by Sun and co-workers explores the robustness of structural superlubricity under pressure load. Experimental data shows that the wearless feature of ss-lubricity is preserved even under GPa-pressures. First-principles calculations clarify the mechanism of interfacial interaction at the ss-lubricity contact, and verify the strong resistance to interlayer bonding under pressure. The work highlights the intrinsic nature of a wearless contact, which is expected for ss-lubricity contacts of weakly-interacting via van der Waals and/or partly-electrostatic interfaces. Moreover, the robustness up to GPa-pressures indicates that the contact can sustain extreme mechanical loads, even higher than the strength of materials of the contact pairs. The work thus appears of high potential importance for both scientific understandings of superlubricity and its engineering applications. I would thus like to recommend it for publication after the authors address my following comments.

1. Superlubricity is usually defined by the frictional characteristics such as the frictional resistance

or frictional coefficient. However, the current work is focused on its wearless feature. It is thus interesting to ask that, even if the wearless feature remains robust, how would the frictional resistance change with the load. From the potential energy landscape perspective, atoms at the contact could be pushed to the 'wells' in the potential energy surface, and the shear stress is expected to increase. Moreover, for incommensurate surfaces what is often of concern for superlubricity studies is that local commensurability could be promoted. Then the local interface registry can further elevate the frictional resistance. Some discussions based on theoretical calculations would be helpful to clarify this issue, and, to some extent, the definition of (structural) superlubricity.

2. One of the most significant findings here is the GPa-pressure the contact can sustain. However, the pressure is estimated from the force loading in experiments through finite element analysis. Although atomic-level calculations further verify this argument, it is still interesting to ask how the mechanism works in experiments. Specifically, at the ss-lubricity contact prepared by stacking two materials with atomically-smooth surfaces, small intercalant-molecules may not be avoided. Wear at the defective sites during sliding at the contact can also result in deposition of debris. The local pressure can then increase, and this effect should be discussed, preferably in a quantitative way.

3. The work suggests that wear can nucleate at defective sites at much reduced pressure. The reason is that the "in-plane" strength of the materials of contact pairs is lower than the force transmitted to the lattice via interfacial sliding. This deserves a discussion, to relate this understanding to the conventional picture of wear (between microscopic rough surfaces). There are similarity and difference between the rough surfaces and the atomically smooth ones. Comparative comments on this aspect may allow to picturize the wear process at ss-lubricity contacts and develop models applicable to material screening.

4. I gather the tests are performed at room T and relatively low speed ($\sim 10^4$ nm/s), yet the energy dissipation/damping in phonons/heat clearly play role at higher speeds (e.g. explored in old MD study by M. Luo et al. *J. Appl. Mech.*, 80, 040906, 2013). Briefly compare the regimes, and comment broader on possible role of T and speed, perhaps important upper bounds to practical high-speed ss-lubricity devices.

Very nice study combining good experiments with balanced in-depth theoretical modeling.

RESPONSE TO REVIEWERS' COMMENTS

Reviewer #1

In this manuscript, the authors present a hybrid study that combines both experimental and theoretical approaches to explore the outstanding robustness of SSL under extremely high pressures up to the pressure limit. The paper thoroughly investigates both the symmetric vdW contact between graphite with incommensurate interfaces and the asymmetric vdW contact between a tungsten tip and graphite substrate. These results provide valuable insight for design SSL with enhanced pressure-resistance. Overall, this research is quite interesting and useful for the readers from the tribology community. There are some issues that require further clarification and explanations before publishing the paper.

Response: Thank you for your insightful comments and kind suggestions. We improved our work accordingly. Please find below our point-to-point responses (in blue) to your comments (in black).

1. The atomically smooth contacts of SSL studied in this work markedly differ from traditional rough contacts. This contrast might account for why the pressure-resistance of common rough contacts falls short of the SSL contacts. Please discuss the difference of pressure-resistance in these contacts.

Response: Thank you for this comment. The pressure of rough contacts with multi-asperities is non-uniform, in which the local maximum pressure, i.e., pressure concentration, limits the pressure-resistance. In contrast, the SSL contacts are atomically smooth with uniform stress distribution, which significantly excludes the influence of rough asperities. This contrast accounts for the ultrahigh pressure-resistance of SSL contacts. In response, we added a discussion in the main text.

Revisions made on page 7 of the main text:

It is noted that different from the conventional rough contacts with multi-asperities^{46,47}, the SSL contacts we studied are atomically smooth and uniform, which can be described as a single-contact model. Without pressure concentration, the wear of the SSL contacts can be directly determined by the interfacial bonding pressure (stage I).

2. The authors investigated the correlation of the breakdown pressure with bulk modulus. This is a quite interesting, since a bulk property is linked with an interfacial property. However, more explanations should be provided for better understanding this behavior.

Response: Thank you for this suggestion. It is known that a high bulk modulus results from a high valence

electron density¹. Meanwhile, the high valence electron density would yield high surface electron density that increases the energy barrier reacted from vdW to covalent contacts with graphite (**Fig. R1**), which accounts for the improvement of the breakdown pressure. The correlation among bulk modulus, valence electron density, surface electron density, and energy barrier helps to understand this behavior. In response, we added an explanation into the main text.

Figure R1 | Relationship between surface valence electron density and valence electron density. The valence electron density and surface valence electron density are calculated using the formulae $\rho = \zeta/V$ and $\rho_s = \zeta/A$, respectively, where ζ , V , and A represent the valence electron¹, atomic volume², and atomic surface area. Herein, the atomic surface area is the surface area occupied per atom.

Revisions made on page 10 of the main text:

The correlation of the breakdown pressure with B can be explained in the following analyses. It is known that metals with a high bulk modulus result from a high valence electron density that corresponds to a high surface electron density⁵⁵. The high surface electron density contributes to an increase in the energy barrier reacted from vdW to covalent contacts with graphite, thereby leading to an improvement in the breakdown pressure. Meanwhile, the strong correlation between the breakdown pressure and IE is analyzed.

3. In many contacts, the materials involved are nearly isotropic, whereas the graphite used in SSL contacts is significantly anisotropic. How does this anisotropy affect the pressure-resistance and its wear behaviors?

Response: Thank you for this comment. The anisotropy effect of graphite can be understood as follows:

Regarding the anisotropy effect on the pressure-resistance, the anisotropy endows graphite with atomically

smooth surfaces, rather than common rough and multi-asperity surfaces. This atomically smooth surface of graphite can form chemically inert vdW contacts with common materials. The ultrahigh pressure-resistance of SSL contacts originates from the transition from vdW contacts to covalent contacts.

Regarding the anisotropy effect on the wear behaviors, the anisotropy endows graphite with ultrahigh in-plane stiffness while ultra-low out-of-plane interactions, which inhibits the in-plane deformation and failure³. Consequently, the incommensurate contact with ultralow friction can be well maintained, which leads to wear-free behaviors.

4. The reported breakdown pressure of SSL, reaching up to several GPa, exceeds the actual compressive strength of many materials with defects. Accordingly, a discussion regarding the influence of material defects on the pressure resistance of SSL contacts would be beneficial.

Response: Thank you for this insightful suggestion. The breakdown mode of SSL under pressure is determined by the weakest one, i.e., the minimum value of the uniaxial compressive strength of materials (metals or graphite) and the interfacial breakdown pressure. Our first-principles calculations indicate that the uniaxial compressive strengths of defect-free tungsten and graphite are as high as 66 GPa and 174 GPa, respectively (**Fig. 4f**). In comparison, experimental results indicate the compressive strength of about 4~14 GPa for the tungsten with grain sizes from 10 nm to 3 μm ⁴, and a phase transition of the graphite under a pressure of 27 GPa⁵. The differences between simulations and experiments reflect the influence of defects on the compressive strength of materials. Besides, we have demonstrated that the interfacial breakdown pressure can be reduced and improved by the vacancy and oxide defects (**Supplementary Notes 6-7**), respectively. These results suggest that the breakdown mode of SSL under pressure can be modified by material defects.

Revisions made on page 10 of the main text:

The destruction of SSL contact is thus a result of the combined effects of structural distortion in the metals and BO change or charge transfer at the interface, which is determined by the weakest one, i.e., the minimum value of the uniaxial compressive strength of materials (metals or graphite) and the interfacial breakdown pressure.

5. If the pressure exceeds the breakdown pressure and is subsequently removed, is it possible for the covalent interface recovered to vdW contacts?

Response: Thank you for this comment. In response, we conducted additional calculations to investigate whether the covalent interface can recover to vdW contacts after removing the pressure. Our calculations demonstrate that the interface of graphite maintains vdW interactions, while the contact between graphite and tungsten maintains

covalent interactions after removing the pressure that exceeds the breakdown pressure (**Fig. S7**). Therefore, the covalent interface would not recover to vdW contacts without other external loads.

Revisions made on page 7 of the main text:

Furthermore, first-principles calculations demonstrate that the interface of graphite maintains vdW interactions, while the contact between graphite and tungsten maintains covalent interactions after removing the external pressure (**Fig. S7a**), and a maximum stress of about 7 GPa is needed to separate the covalent contact (**Fig. S7b**). These results suggest a stable covalent contact formed between graphite and tungsten.

Figure S7 | Stability of the covalent contact between graphite and tungsten. a, Structures before and after removing the pressure that exceeds the breakdown pressure. **b**, Uniaxial separation tests of the covalent contact, which indicate that a maximum stress of about 7 GPa is needed to separate the covalent contact.

6. More details on the experimental determination of the breakdown pressure should be provided.

Response: Thank you for this comment. The experimental determination of the breakdown pressure follows three steps: (1) Determine the critical load beyond which rupture of materials can be first observed (see **experiments in the main text**). (2) Characterize geometries and material properties of the contact (see **Supplementary Note 1: Determination of the contact pressure**). (3) Calculate the critical pressure under the critical load by using the finite element analysis (see **Supplementary Note 1: Determination of the contact pressure**). In response, we added more details into the manuscript.

Revisions made on pages 3, 5, and 6 of the main text:

Figure 1c, d shows the OM and AFM images of the graphite substrate after a sliding test under the load of 10

mN (the highest accessible normal load by the loading system which corresponds to the contact pressure of 9.45 GPa, see **Supplementary Note 1** for the determination of the pressure) and a sliding distance up to 10 mm in 5×10^2 cycles.

Figure 2c, d shows the OM and AFM images of the graphite substrate after sliding tests under loads of 0.2 mN and 0.5 mN (corresponds to the pressure of 3.74 GPa and 5.07 GPa, see **Supplementary Note 1** for the determination of the pressure), respectively, with a sliding velocity of 10 $\mu\text{m/s}$ over a reciprocating amplitude of 30 μm .

The breakdown pressure of the tungsten/graphite contact is higher than 3.74 GPa, beyond which wear was initially observed.

Revisions made on pages 2 and 6 of Supplementary Note 1: Determination of the contact pressure:

1.1 Graphite/graphite contact

Contact pressure at the graphite/graphite contact was calculated using the finite element analysis (FEA) (**Fig. S1a**). ... The model was constructed in consistency with the experimental setup, where the radius of the tungsten tip is 3.5 μm and the thicknesses of the graphite mesa and the mechanically exfoliated graphite substrate are 150 nm and 20 nm, respectively (**Fig. S2**). **Figure S3b** presents a contour of the pressure distribution at the graphite/graphite contact under the load of 10 mN. The critical pressure is valued as the peak stress (9.45 GPa) at the contact. The effective contact radius is 0.8 μm (**Fig. S3a**).

1.2 Tungsten/graphite contact

Contact pressure at the tungsten/graphite contact was also calculated using FEA (**Fig. S4a**). Material parameters remain the same as those for the graphite/graphite contact. ... The radius of the tungsten tip is 1.75 μm (**Fig. S8**) and the thicknesses of the graphite layer and the silica film are 10 nm and 300 nm, respectively. The critical pressure is valued as the peak stress (3.74 GPa, **Fig. S4c**) at the contact under the experimentally determined critical load (0.2 mN). The effective contact radius is 0.16 μm (**Fig. S4e**).

Reviewer #2

The paper describes high pressure experiments of layered materials. The authors find that superlubricity conditions can be maintained up to 9GPa in the case of graphite mesa on graphite and 3GPa for tungsten tip on graphite. The analysis of a variety of metals on graphite points to the influence of the bulk modulus and the ionization energy to be the most important parameters to have a high threshold for the transition to wear conditions. Below the threshold the authors report wearless friction. This is a quite important observation, which differs from the more common wear models, which assume a continuous removal of atoms from the wear zone. Therefore, superlubricity conditions seem to be suitable to drastically reduced wear at relatively high pressures. In summary, I support publication of this important observation.

Response: We appreciate your insightful comments and kind suggestions. We improved our work accordingly. Please find below our point-to-point responses (in blue) to your comments (in black). The revisions are shown in blue color in the revised manuscript.

Minor points:

-It would be interesting to compare the observed absence of wear with typical wear rates under similar conditions of more common materials.

Response: Thank you for the suggestion. We added a comparison of typical wear rates in common materials to the observed absence of wear in the graphitic SSL.

Revisions made on page 4 of the main text:

The resolution of the AFM characterization in **Fig. 1d** is $0.1 \mu\text{m} \times 0.1 \mu\text{m} \times 1 \text{ nm}$, which corresponds to the absence of wear below the minimum detectable wear rate of $10^{-10} \text{ mm}^3/\text{N m}$ under the load of 10 mN and the sliding distance of 10 mm. In comparison, the wear rates of the industrial materials of steel/steel and the microscopic materials of silicon/silicon nitride are $10^{-7} \sim 10^{-3} \text{ mm}^3/\text{N m}^{43}$ and $10^{-6} \sim 10^{-4} \text{ mm}^3/\text{N m}^{44}$, respectively. The result demonstrates the exceptional wear resistance of SSL.

-It might be also of interest to explore the temperature stability of superlubricity conditions. Some comments on this point might be of interest to the reader.

Response: Thank you for the suggestion. In response, we provided outlooks on the temperature stability of superlubricity conditions in the main text.

Revisions made on page 11 of the main text:

Temperature is a crucial factor in practical applications. The temperature stability of SSL can be comprehended from two aspects. On the one hand, high temperature can help overcome the energy barrier during sliding and result in reduced friction^{56,57}. On the other hand, the breakdown process of SSL can be regarded as a chemical reaction^{28,29}. The energy barrier of such a reaction can be more easily overcome under a higher temperature, which consequently reduces the pressure-resistance of SSL.

-It would be also interesting to comment on the influence of humidity.

Response: Thank you for the comment. In response, we provided comments on the influence of humidity in the main text.

Revisions made on page 11 of the main text:

Humidity is another important factor to be considered. Considering the squeeze-out effect at the hydrophobic surface of graphite, the effect of adsorbed water is negligible. This is verified by previous work which reported no obvious difference in friction of SSL in the mesa/substrate setup under ambient conditions (relative humidity of 42%) and nitrogen atmosphere (relative humidity < 10%)²², indicating the stability of SSL with the change of humidity.

Reviewer #3

Sun et al. present a study combining experiments and atomistic modeling, in which they find low friction in two systems to remain robust up to Gigapascal stresses. Since others have found robust superlubricity in similar systems, e.g., Ref. 17, 20, 21, and 22, I find the reported progress somewhat incremental. In fact, Web Of Science identifies 18 papers with the keyword "robust superlubricity" in the title and most of them are concerned with graphite and/or graphene-terminated surfaces under a relatively large compressive stress, just like the submitted manuscript. Thus, the robustness of superlubricity is no news.

Response: Many thanks to the reviewer for providing insightful comments. To be brief, we reported that the performance of structural superlubricity (SSL) can be wear-free up to several GPa, well beyond the compressive strength of many materials. This finding is of vital importance for SSL and related applications, and has NOT been discussed in any of the publications mentioned in the comment.

We improved our work according to your suggestions. Specifically, the major novelty of our work is the observed wear behavior and uncovered wear mechanism of SSL induced by pressure-assisted bonding, rather than demonstrating the low friction under relatively large compressive stress in previous works. The breakdown mechanism is crucial for the engineering application of SSL but somehow has got little attention in previous works. In our work, we first experimentally observed the long-distance wear-free feature of the graphite/graphite and tungsten/graphite contacts under ultrahigh pressures of 9.45 GPa and 3.74 GPa, respectively, demonstrating the robustness of SSL under ultrahigh pressure. Following that, by increasing the pressure further, we found a critical pressure beyond which deformation and failure of the graphite initiate. This behavior contrasts significantly with the conventional picture of wear, which typically involves wear accumulation over the loading time^{6,7}. The breakdown of SSL was explained by a step-wear mechanism following the experimental evidence to include (I) wear initiation through pressure-assisted interfacial bonding, and (II) wear development through shear-induced tearing of the graphite layers. To our knowledge, there is no prior paper that actually reported such wear behavior and mechanism of SSL contacts. We hope this explanation helps the reviewer to better understand our work.

The potential novelty of this paper is the attempt to rationalize the breakdown of graphite's superlubricity. However, I find the connection between the DFT calculations and the experiments rather weak. What the authors appear to find is a hybridization instability of carbon atoms from sp² to sp³ when being squished against a metal, which could be perceived as an adhesive buckling instability. However, the authors failed to investigate if this

instability is reversible when the normal stress is removed. If they did observe mass transfer under decompression, they could have made a stronger case that the simulations pertain to the experiments. This way, the authors only find an instability, which, however may very well be elastic in nature.

Response: Thank you for this comment. Regarding the connection between the DFT calculations and the experiments, the explanations are as follows: In our experiments, the contact is atomically smooth with a uniform pressure distribution at the microscale. Consequently, the experimental contact model can be reasonably simplified to the model used in DFT calculations. Regarding the reversibility of the contact when the normal stress is removed, we conducted additional calculations. Our calculations demonstrate that the contact between graphite and tungsten maintains covalent interactions after removing the pressure (**Fig. S7a**), and a maximum stress of about 7 GPa is needed to separate the covalent contact (**Fig. S7b**). These results indicate that the transition triggered by pressure is stable, which is NOT elastic in nature.

The shear strength of the interface increases dramatically with pressure-assisted bonding (**Fig. 3d**), which indicates the breakdown of SSL. We then conducted interfacial sliding using molecular dynamics (MD) simulations with parameters determined by first-principles calculations. The MD simulations captured the failure of graphene (mass transfer) during the sliding process (**Fig. 3f**, see also **Supplementary Movie S3**), and explained the experimental observations of wear in the form of tearing of graphite layers. These simulations support the mass transfer under the sliding and provide a case that the simulations pertain to the experiments.

Revisions made on page 7 of the main text:

Furthermore, first-principles calculations demonstrate that the interface of graphite maintains vdW interactions, while the contact between graphite and tungsten maintains covalent interactions after removing the external pressure (**Fig. S7a**), and a maximum stress of about 7 GPa is needed to separate the covalent contact (**Fig. S7b**). These results suggest a stable covalent contact formed between graphite and tungsten.

Figure S7 | Stability of the covalent contact between graphite and tungsten. a, Structures before and after removing the pressure that exceeds the breakdown pressure. **b,** Uniaxial separation tests of the covalent contact, which indicate that a maximum stress of about 7 GPa is needed to separate the covalent contact.

What I found truly disappointing is that this hybridization induced buckling instability, which is a purely quantum mechanical effect, was related to the well-known puckering effect, which is related to the classical, thin-sheet mechanics of a graphite layer. I see little relation between the two. The one aspect that does make sense, however, is that the onset of buckling is related to the question how readily graphite's counter face is donates electrons to graphite.

Response: Thank you for this comment. Interatomic interaction, either bonding (e.g., covalent, metallic, ionic) or non-bonding (e.g., van der Waals, electrostatic), originates from the quantum mechanics of electrons, which is crucial for the homo/hetero-interfaces studied here under pressure. There might be a misunderstanding that the hybridization-induced buckling instability was related to the puckering effect. The buckling instability in this work is displayed during the compression process (**Fig. 3c**, first-principles calculations) because of the lattice mismatch between the tungsten and the graphite layer (with bond lengths perturbed by the interaction with tungsten) in the reduced DFT model with an enforced periodic boundary condition. The bond length of sp^3 hybridized carbon (e.g., 0.154 nm for diamond) is longer than that of sp^2 (e.g., 0.142 nm for graphite). This fact is recognized by the reviewer - the electrons donated by the counter face of graphite lead to the onset of buckling, which is also supported by our simulations showing charge transfer across the interface. For large-scale contact in experiments that our DFT calculations cannot handle, the residual strain induced by hybridization can be released by reducing the areal density of pressure-induced hybridization. However, the puckering effect shown

during the sliding process (**Fig. 3f**, MD simulations) at the edge of contact is due to the accumulated in-plane deformation of the graphite layer not in contact, which prefers to bend instead of being compressed considering the weak van der Waals cohesion at the interface. The puckering effect is thus absent in models with periodic boundary conditions. However, for a finite contact, the puckering effect becomes more significant if the load transfer into the graphite increases as suggested by our MD simulations (**Fig. 3f**), which could be triggered by the pressure-induced hybridization. The key information fed into the MD simulations includes the interaction strength at the interface, which was obtained from the DFT calculations. We clarified these points in the revision.

Revisions made on page 7 of the main text:

This conclusion is validated by our MD simulations (see **Supplementary Movie S3**) with the key information including the interaction strength at the interface parameterized by the DFT calculations.

Revisions made on page 8 of the main text:

Figure 3 | c, Snapshots as the tungsten is forced to move towards graphite, which shows the structural responses under pressure. The decline of pressure is attributed to the formation of pressure-assisted interfacial bonds. The buckling behavior at contact is triggered by the lattice mismatch between the tungsten and the graphite layer in the reduced DFT model with an enforced periodic boundary condition. **d**, Shear stress-displacement relation below and beyond the P_{cr} , obtained from DFT calculations. **e**, Snapshots of the sliding of tungsten on graphite after the formation of interfacial bonds. The sliding process demonstrates the transformation of sp^3 hybridization state of carbon atoms, including the break, formation, and movement of W-C bonds. **f**, Illustration of the wear process. The puckering effect at the front of contact is due to the accumulated in-plane deformation of the graphite layer not in contact, which prefers to bend instead of being compressed considering the weak vdW cohesion at the interface.

The part that I found the most disappointing, however, is that the friction reported from DFT is computed only over a sliding distance of 1 Angstrom. To get reliable numbers, shear forces must be averaged over periodic patterns, e.g., multiple integers of the lattice constant, or, in disordered or symmetry broken systems, over a meaningfully long sliding distance. The friction force reported in Fig. 3d is about to become negative so that it is absolutely impossible to even crudely estimate a mean frictional stress from that graph. It would be very interesting to see how the sp^3 hybridization state of carbon atoms does get dragged along and if/how it progresses to the next atom. If there were instabilities, then the breakdown of superlubricity could be argued to arise from the sp^2 to sp^3 hybridization changes, but the relation to wear would nonetheless still be missing.

Response: Thank you for this comment. We realized that the previous expression of ‘frictional stress’ is inappropriate since we did static shear tests. Consequently, we revised the expression of ‘frictional stress’ to ‘shear stress’. Friction, instead of shear, should be measured as the resistance to motion or the derivative of energy with sliding distance, which includes all channels of energy dissipation (e.g., phononic or electronic excitation, plasticity, wear). Using first-principles shear tests, we compared the shear behavior before and after the breakdown pressure. The shear stress calculated by first-principles calculations reflects the static friction, different from the kinetic friction in experiments. To address the reviewer’s concerns, we extended the sliding distance over a period (**Fig. R2**), which demonstrates that the peak shear stress in **Fig. 3d** is the shear strength during the shear process. Furthermore, we replotted **Fig. 3e** to demonstrate the transformation of sp^3 hybridization state of carbon atoms, including the break, formation, and movement of W-C bonds. This accounts for the high shear strength of the covalent contact. Finally, the wear process through shear-induced wrinkling and tearing of graphite layers was explored by MD simulations, in which the parameters were determined by the above first-principles

shear simulations. It should be mentioned that in MD simulations with large sliding distances, the mechanical energy dissipation processes can be captured, which can measure the friction, while our shear stress responses in the DFT calculations only provide part of the information – the energy released in the negative stress region is usually dissipated into phonons or heat and cannot be recovered. We added this discussion to the revision.

Figure R2 | Shear stress-displacement relation before and after P_{cr} over a period.

Revisions made on page 7 of the main text:

It should be mentioned that our shear stress responses in the DFT calculations only provide part of the information, different from typical molecular dynamics (MD) simulations with large sliding distances which were adopted to capture the mechanical energy dissipation processes and measure the friction.

Revisions made on page 8 of the main text:

Figure 3 | d, Shear stress-displacement relation below and beyond the P_{cr} , obtained from DFT calculations. e,

Snapshots of the sliding of tungsten on graphite after the formation of interfacial bonds. The sliding process demonstrates the transformation of sp^3 hybridization state of carbon atoms, including the break, formation, and movement of W-C bonds.

To make the case for wear, one must study the contact edges, in particular the stress distribution near the trailing edge, where tensile stresses are largest. Such an analysis was not presented in the given work and requires advanced modeling skills.

Response: Thank you for this comment. Indeed, the tensile stress near the trailing edge is the largest, and our MD simulations demonstrate a failure near the trailing edge (**Fig. 3f**). This is because sliding the tungsten tip will lead to the movement of carbon atoms inside the graphite layer due to the covalent-bonding interfaces. Therefore, the in-plane deformation of graphite layer accumulates and finally leads to the failure near the trailing edge (**Fig. R3**). In addition, we have also provided analyses according to the shear-lag model. We estimated a critical size (l_c) of effective contact and proposed two different modes of failure or deformation. As the size of contact is larger than l_c , graphene will be torn by shear-induced tension in the basal planes, whereas the tungsten-carbon bonds break and the interfacial load transfer is insufficient as the size of contact is below l_c . This ‘advanced’ multiscale model (interaction strength at the interface obtained from DFT calculations, load transfer and wear behavior simulated by MD, see **Fig. 3f** for details) provides the essential physics and the stress information (through strain) mentioned in the comment.

Figure R3 | Snapshots of the sliding of tungsten on graphite after the formation of interfacial bonds. The atoms are colored according to the in-plane tensile strain.

Revisions made on page 7 of the main text:

Using the shear-lag model (see **Supplementary Note 5** for details), we estimated that the transverse load at the interface with pressure-assisted bonds can be transferred into graphite upon interfacial sliding, which can damage the graphite layers with a high in-plane tensile strength⁴⁹ and activate wear^{50,51} at the interface. ...The deformation and failure of graphite layer near the trailing edge (**Fig. 3f**) indicates the peak stress at the edge, which agrees well with previous shear-lag model prediction⁵².

In conclusion, I do not see the manuscript to meet the criteria of Nature Communications. The experimental results are sufficiently interesting to be disseminated in a good, regular journal, while the analysis of the simulation results (which certainly are interesting) would require a much cleaner analysis of the observed instability and its effect on friction and wear.

Response: Thank you for your comments. To make our analysis clearer to the reader, we added more explanations and discussions following your suggestions to the revision. We hope our responses and revisions adequately address your concerns and offer a better understanding of our work.

Reviewer #4

The report by Sun and co-workers explores the robustness of structural superlubricity under pressure load. Experimental data shows that the wearless feature of ss-lubricity is preserved even under GPa-pressures. First-principles calculations clarify the mechanism of interfacial interaction at the ss-lubricity contact, and verify the strong resistance to interlayer bonding under pressure. The work highlights the intrinsic nature of a wearless contact, which is expected for ss-lubricity contacts of weakly-interacting via van der Waals and/or partly-electrostatic interfaces. Moreover, the robustness up to GPa-pressures indicates that the contact can sustain extreme mechanical loads, even higher than the strength of materials of the contact pairs. The work thus appears of high potential importance for both scientific understandings of superlubricity and its engineering applications. I would thus like to recommend it for publication after the authors address my following comments.

Response: Thank you very much for your recommendation and valuable comments. We improved our work accordingly. Please find below our point-to-point responses (in blue) to your comments (in black).

1. Superlubricity is usually defined by the frictional characteristics such as the frictional resistance or frictional coefficient. However, the current work is focused on its wearless feature. It is thus interesting to ask that, even if the wearless feature remains robust, how would the frictional resistance change with the load. From the potential energy landscape perspective, atoms at the contact could be pushed to the ‘wells’ in the potential energy surface, and the shear stress is expected to increase. Moreover, for incommensurate surfaces what is often of concern for superlubricity studies is that local commensurability could be promoted. Then the local interface registry can further elevate the frictional resistance. Some discussions based on theoretical calculations would be helpful to clarify this issue, and, to some extent, the definition of (structural) superlubricity.

Response: Thank you for this interesting comment and valuable suggestion. We divided this comment into three parts so that our response can be more clearly understood.

First, regarding the comment on the frictional resistance under load: As the reviewer recognized, the shear stress at the sliding interface could be affected by normal pressure at the microscale because of the lattice deformation. However, the in-plane deformation of graphite under high pressure is negligible due to its ultrahigh in-plane stiffness. This is verified by our experimental results which demonstrate that the friction coefficients (the local derivative of the friction force with respect to the applied normal load) of the graphite/graphite contact and the tungsten/graphite contact are in the order of 10^{-5} (Fig. R4) and 10^{-3} (Fig. S6) under the gigapascal pressure, respectively. In addition, we have conducted DFT calculations on the shear behavior of the tungsten/graphite

contact under various pressures. These results also demonstrate the ultralow shear resistance in the tungsten/graphite contact below the breakdown pressure (**Fig. R5**). In response, we added the discussion in the main text.

Second, regarding the concern on the definition of structural superlubricity: Theoretically, structural superlubricity (SSL) is defined as a behavior where no energy dissipation exists between two sliding surfaces⁸⁻¹⁰. The SSL has been verified by experiments which show the very small kinetic friction resistance¹¹ and the sub-linear dependence of friction force F on the contact area A ($F \propto A^\alpha$, $\alpha \leq 1$)^{12,13}. In the engineering view, the state of SSL is usually defined by the sliding contact with a friction coefficient on the order of 10^{-3} ¹⁴. All the experimental results of the studied SSL contacts meet the definition.

Third, regarding the local interface registry: The buckling behavior, which may trigger the in-plane deformation and change the local commensurability, is displayed during the compression process because of the lattice mismatch between the tungsten and the graphite layer. However, for large-scale contact in experiments that our DFT calculations cannot handle, the residual strain induced by hybridization can be released by reducing the areal density of pressure-induced hybridization, which will not induce the change of local interface registry.

Figure R4 | Experimental measurement of the friction-load relation of the graphite/graphite contact.

Figure S6 | Experimental measurement of the friction-load relation of the tungsten/graphite contact.

Figure R5 | DFT calculation results on shear stress-displacement relation of the tungsten/graphite contact under various pressures.

Revisions made on page 7 of the main text:

Before the formation of interfacial W-C bonds, the shear strengths calculated by first-principles simulations are maintained in the low amplitude state (0.12, 0.09, and 0.11 GPa under the pressure of 0.8, 2.4, and 3.4 GPa, respectively), demonstrating the ultralow shear resistance at the tungsten/graphite contact below the breakdown pressure.

2. One of the most significant findings here is the GPa-pressure the contact can sustain. However, the pressure is estimated from the force loading in experiments through finite element analysis. Although atomic-level

calculations further verify this argument, it is still interesting to ask how the mechanism works in experiments. Specifically, at the ss-lubricity contact prepared by stacking two materials with atomically-smooth surfaces, small intercalant-molecules may not be avoided. Wear at the defective sites during sliding at the contact can also result in deposition of debris. The local pressure can then increase, and this effect should be discussed, preferably in a quantitative way.

Response: Thank you for the concern and suggestion. Additional discussions on intercalant molecules, inserted defects, and debris are provided below:

Regarding the intercalant molecules which are unavoidable during sample preparations in experiments, we carefully designed experimental procedures where a pre-cleaning step was conducted in a large area of the substrate before tests were carried out in the smaller central region (**Fig. 1c**). During the cleaning, friction at the assembled mesa/substrate contacts decreased with the increasing sliding cycles and ended into a stable state after about 15 cycles (**Fig. R6**), demonstrating the removal of the confined intercalant molecules at contact¹⁵.

Regarding the inserted defects inside the sliding interface, we have studied cases of vacancy and oxide defects. The results demonstrate that the interfacial breakdown pressure can be reduced and improved by the vacancy and oxide defects (**Supplementary Notes 6-7**), respectively. The elevated breakdown pressure caused by oxide defects originates from the enhanced chemical inertness, ensuring the ultrahigh pressure-resistance of SSL. The above analyses suggest that the wear resistance of SSL can be tuned by defects.

Regarding the deposition of debris: The experimental result (**Fig. 1d**) displayed debris at the boundary of the pre-cleaned region, whereas neither aggregation of debris nor rupture of the material was observed in the sliding-test region. This result suggests that the debris was created by removing contaminants inside the interface during the pre-cleaning step and presents the wear-free feature of the studied SSL contacts.

Accordingly, we added discussions in the main text.

Figure R6 | Self-cleaning effect in the mesa/substrate contact in graphite.

Revisions made on page 12 of the main text:

Finally, for the intercalant molecules which are unavoidable during sample preparations, the pre-cleaning step before tests can remove the confined molecules. The influence of the intercalant molecules is studied by simulations (**Supplementary Notes 6-7**), which demonstrate that the interfacial breakdown pressure can be tuned by defects and the ultrahigh pressure-resistance of SSL can be well maintained by enhancing the chemical inertness at contacts.

3. The work suggests that wear can nucleate at defective sites at much reduced pressure. The reason is that the “in-plane” strength of the materials of contact pairs is lower than the force transmitted to the lattice via interfacial sliding. This deserves a discussion, to relate this understanding to the conventional picture of wear (between microscopic rough surfaces). There are similarity and difference between the rough surfaces and the atomically smooth ones. Comparative comments on this aspect may allow to picturize the wear process at ss-lubricity contacts and develop models applicable to material screening.

Response: Thank you for the comment. In general, wear can be determined by two different mechanisms, including the physical mechanism which refers to surface roughness, and the chemical mechanism which describes reactivity at the contact interface. The conventional picture of wear for the rough contact is affected by the combination of both the physical and chemical mechanisms, where the fracture of sharp asperities and the bond transition at the local maximum pressure dominate the pressure-resistance, presenting a feature of gradual wear^{6,7,16,17}. In other words, the wear of rough contacts is usually determined by the deformation IN the material instead of at the interface. In contrast, the SSL contacts we studied are atomically smooth with uniform stress distribution, where the wear of the SSL contacts can be ideally determined by the interfacial bonding pressure (chemical mechanism), below which no damage is activated and accumulated. In response, we added discussions in the main text.

Revisions made on page 7 of the main text:

It is noted that different from the conventional rough contacts with multi-asperities^{46,47}, the SSL contacts we studied are atomically smooth and uniform, which can be described as a single-contact model. Without pressure concentration, the wear of the SSL contacts can be directly determined by the interfacial bonding pressure (stage I).

4. I gather the tests are performed at room T and relatively low speed ($\sim 10^4$ nm/s), yet the energy dissipation/damping in phonons/heat clearly play role at higher speeds (e.g. explored in old MD study by M. Luo

et al. J. Appl. Mech., 80, 040906, 2013). Briefly compare the regimes, and comment broader on possible role of T and speed, perhaps important upper bounds to practical high-speed ss-lubricity devices.

Response: Thank you for the insightful comment. Based on theoretical calculations, the speed-friction relation can be divided into three regimes: thermolubric, stick-slip, and ballistic (**Fig. R7a**)¹⁸. Experiments conducted by common frictional tests only covered a limited range^{19,20}, which falls in the stick-slip regime showing an approximate positive linear relationship (**Fig. R7b**)²¹. In response, more comments on the high-speed regime and the role of temperature and speed are provided in the main text.

Figure R7 | a, Three friction regimes of a pinned nanoslider as a function of velocity: thermolubric, stick-slip, and ballistic¹⁸. **b**, Comparison between the experimental results and simulation results²¹.

Revisions made on pages 11 and 12 of the main text:

Temperature is a crucial factor in practical applications. The temperature stability of SSL can be comprehended from two aspects. On the one hand, high temperature can help overcome the energy barrier during sliding and result in reduced friction^{56,57}. On the other hand, the breakdown process of SSL can be regarded as a chemical reaction^{28,29}. The energy barrier of such a reaction can be more easily overcome under a higher temperature, which consequently reduces the pressure-resistance of SSL.

Besides, the speed range covered by common friction tests conducted by the AFM and tribometers is 10^{-7} ~ 10^{-2} m/s^{3,22}. Under higher speeds, the enhanced energy dissipation may hinder applications of SSL. However, some interesting discoveries suggest that SSL can be sustained under the speed from 25 m/s¹⁹ to 294 m/s⁵⁸, which sheds light on applications of SSL in the ultrahigh-speed regime, such as gigahertz oscillators⁵⁹.

Very nice study combining good experiments with balanced in-depth theoretical modeling.

Response: Thank you for your recognition of our work. We hope our responses and revisions satisfied your

concern.

REFERENCES

1. Economou, E. N. *The Physics of Solids*. (Springer Berlin, Heidelberg, 2010).
2. Jin, R., Yuan, X. & Gao, E. Atomic stiffness for bulk modulus prediction and high-throughput screening of ultraincompressible crystals. *Nat. Commun.* **14**, 4258 (2023).
3. Lee, C., Wei, X., Kysar, J. W. & Hone, J. Measurement of the elastic properties and intrinsic strength of monolayer graphene. *Science* **321**, 385-388 (2008).
4. Yang, J. *et al.* Strength enhancement of nanocrystalline tungsten under high pressure. *Matter Radiat. Extrem.* **5**, 058401 (2020).
5. Ke, F. *et al.* Synthesis of atomically thin hexagonal diamond with compression. *Nano Lett.* **20**, 5916-5921 (2020).
6. Jia, K. & Fischer, T. E. Sliding wear of conventional and nanostructured cemented carbides. *Wear* **203**, 310-318 (1997).
7. Chung, K. H. & Kim, D. E. Fundamental investigation of micro wear rate using an atomic force microscope. *Tribol. Lett.* **15**, 135-144 (2003).
8. Hirano, M. & Shinjo, K. Atomistic locking and friction. *Phys. Rev. B* **41**, 11837-11851 (1990).
9. Hirano, M. & Shinjo, K. Superlubricity and frictional anisotropy. *Wear* **168**, 121-125 (1993).
10. Shinjo, K. & Hirano, M. Dynamics of friction: Superlubric state. *Surf. Sci.* **283**, 473-478 (1993).
11. Dienwiebel, M. *et al.* Superlubricity of graphite. *Phys. Rev. Lett.* **92**, 126101 (2004).
12. Dietzel, D., Feldmann, M., Schwarz, U. D., Fuchs, H. & Schirmeisen, A. Scaling laws of structural lubricity. *Phys. Rev. Lett.* **111**, 235502 (2013).
13. Wang, J. *et al.* Generalized scaling law of structural superlubricity. *Nano Lett.* **19**, 7735-7741 (2019).
14. Peng, D. *et al.* 100 km wear-free sliding achieved by microscale superlubric graphite/DLC heterojunctions under ambient conditions. *Natl. Sci. Rev* **9**, nwab109 (2022).
15. Deng, H., Ma, M., Song, Y. M., He, Q. C. & Zheng, Q. S. Structural superlubricity in graphite flakes assembled under ambient conditions. *Nanoscale* **10**, 14314-14320 (2018).
16. Gotsmann, B. & Lantz, M. A. Atomistic wear in a single asperity sliding contact. *Phys. Rev. Lett.* **101**, 125501 (2008).
17. Jacobs, T. D. & Carpick, R. W. Nanoscale wear as a stress-assisted chemical reaction. *Nat. Nanotechnol.* **8**, 108-112 (2013).
18. Krylov, S. Y. & Frenken, J. W. M. The physics of atomic-scale friction: Basic considerations and open questions. *Phys. Status Solidi B* **251**, 711-736 (2014).
19. Song, Y. *et al.* Robust microscale superlubricity in graphite/hexagonal boron nitride layered heterojunctions. *Nat. Mater.* **17**, 894-899 (2018).
20. Berman, D., Deshmukh, S. A., Sankaranarayanan, S. K., Erdemir, A. & Sumant, A. V. Macroscale superlubricity enabled by graphene nanoscroll formation. *Science* **348**, 1118-1122 (2015).
21. Wang, J., Khosravi, A., Vanossi, A. & Tosatti, E. Sliding and pinning in structurally lubric 2D material interfaces. arXiv:2305.19740 (2023).

REVIEWER COMMENTS

Reviewer #1 (Remarks to the Author):

The authors have responded well to the reviewer's comments, and I recommend that the manuscript be accepted for publication.

Reviewer #2 (Remarks to the Author):

The authors have addressed the points raised by the referees and the manuscript is improved. I recommend to publish this paper in its present form.

Reviewer #3 (Remarks to the Author):

While I appreciate the experiments presented by Sun, Gao, Jia, and coworkers, and see some of my comments addressed, I still do not believe that this paper is a significant step forward in our understanding of superlubricity and/or the onset of wear in graphite or hard materials. Even worse, the more I think about the claimed novelties of the paper, the less I recognize them and the more problems I see regarding the modeling of the paper.

First, correlating ionization energies with stiffness of solids goes as far back as to Slater, though I apologize for not remembering the precise reference. Correlations of bulk and shear modulus are commonly reported in works on interatomic potentials, etc. Second, I am not sure that much can be learned from a theoretical analysis of perfectly flat tungsten without oxide layer squeezed against graphite. Both unavoidable roughness and termination with an oxide layer have an extreme effect on the interaction. Third, the authors fail to deliver the simplicity of the real argument and the reason why graphite has been used as lubricant for more than a century. Its softness in the normal direction makes graphite have relatively small contact pressures (which the authors overcome by using hard tips and localized forces, the strong part of the work) given a rough counterface. Within the plane, bonds are stiff due to the sp² hybridization yielding in-plane moduli in the order of TPa. Since it typically takes say 1% of the Young's modulus (number vary quite a bit and are scale dependent, but to give a rough guess) in Cauchy stress to induce plasticity, the reported numbers make perfect sense, but the explanations are not to the point (as in other cases).

Reviewer #4 (Remarks to the Author):

The authors addressed my comments/questions in quite thorough matter. I also see that the critique from the other reviewers is well considered and is addressed both in the authors' responses and in revisions to the manuscript. This includes also their reasonable argument regarding the degree of novelty raised by the R3. The work establishes a nonlinear critical change in lubricity to friction and wear beyond certain critical pressure, a finding of significance to the field and related fields, perhaps for many other material-pairs and interfaces. Overall, I find the manuscript now in good shape and worthy acceptance for publication in Nat. Comm. in present form.

Reviewer #5 (Remarks to the Author):

I have reviewed the manuscript by Sun et al, along with the supplementary material, the referee reports, the rebuttal, and the subsequent referee correspondence. My view is that this manuscript can be suitable for publication in Nat. Comm. if certain issues can be addressed, which are listed further below. First, regarding overall novelty, the experimental work is novel. SSL has not been shown to be wear-free over large sliding distances at high pressures in ambient conditions. This is

a significant advance and is the key point of novelty of the work. The authors state in response to Reviewer 3 that "To our knowledge, there is no prior paper that actually reported such wear behavior and mechanism of SSL contacts". I agree with this statement.

Regarding Reviewer #3's remaining concerns, I have the following comments:

1. "First, correlating ionization energies with stiffness of solids goes as far back as to Slater, though I apologize for not remembering the precise reference. Correlations of bulk and shear modulus are commonly reported in works on interatomic potentials, etc."

The discussion of this point in the manuscript is in need of revision as I specify below, but is not a reason to reject the paper.

2. "Second, I am not sure that much can be learned from a theoretical analysis of perfectly flat tungsten without oxide layer squeezed against graphite. Both unavoidable roughness and termination with an oxide layer have an extreme effect on the interaction."

The referee has valid points here, but I expect they are addressable. The DFT presented in the paper for clean W is not relevant for the results. They should simulate the contact of tungsten oxide with graphite, not metallic W. The native oxide of W is reported in literature to be ~5 nm thick. I'm not sure of its stoichiometry or structure, but the authors should check the literature or, even better, do surface science on their W (doing XPS on the W material etched with the same KOH procedure should suffice; it's hard to do it on the tip itself although TEM could achieve this) to determine the oxidation state of the W. Then they should simulate this. I suspect they will see that covalent bonds still can form, but probably under even higher pressures since the W atoms will be less likely to bond with C when in higher oxidation states. Defects in the graphite, as the authors show, would lead to lower pressures where wear occurs, in line with experiments. In short, the simulations should match the experiments. The end of the tip is not W, it's WO_x. The O adsorbates simulated in Fig. S13 are not sufficient; they should simulate an oxide, not just a few O adsorbates.

Regarding roughness, this is a universal problem in experiments: even a "smooth" tip will have atomic-scale roughness from steps, defects, grain boundaries, or amorphous regions that will complicate the contact mechanics. However, the graphite is rather compressible in the vertical direction and this may well lead to relatively conformal contact. This is not a reason to reject the paper. If there is literature on the roughness of W tips etched in the same manner as the present manuscript, the authors could refer to that for support (i.e., perhaps others have done high-res SEM or TEM of such etched tips in the past). Alternately the authors could try to take an AFM image of the end of the tip (not easy, but actually done by many groups previously). The optical is not sufficient (Fig. S8) for this; it's a start, as it rules out gross transfer of C to the tip, but more is needed to confirm the tip is reasonably smooth. It could be valuable to take the overall W tip radius and the roughness power spectrum for the W, and use the model of Robbins & Pastewka for rough round tips with self-affine surfaces to get an estimate of the contact pressures. This is not required, but would strengthen the paper.

Regarding contaminants: again, this is a universal problem in experiments and more importantly, such contaminants will be present in typical applications. The cleaning procedure the authors show is sufficient to address this. I recommend they include Fig. R6 from the response to Reviewer 4 in the Supplementary Material to further support their argument about the effectiveness of their cleaning procedure. That said, they need to modify their language about this issue. They surely remove some molecules, but they do not know how much (no one does! It's extremely difficult to measure this). The statement on page 12 that "the pre-cleaning step before tests can remove the confined molecules" should be modified to something like "the pre-cleaning step before tests removes some of the confined molecules, reducing friction to a steady-state level" (and refer to Fig. R6). An additional comment acknowledging that contaminants may still be present is needed, since they cannot rule that out. Finally, the statement "The influence of the intercalant molecules is studied by simulations (Supplementary Notes 6-7)..." needs to be removed; Supplementary Notes 6 and 7 say nothing about contaminant molecules. They only simulate vacancy defects and oxygen adsorbates. The contaminants are likely hydrocarbons and water.

3. "Third, the authors fail to deliver the simplicity of the real argument and the reason why graphite has been used as lubricant for more than a century. Its softness in the normal direction makes graphite have relatively small contact pressures (which the authors overcome by using hard tips and localized forces, the strong part of the work) given a rough counterface. Within the plane, bonds are stiff due to the sp^2 hybridization yielding in-plane moduli in the order of TPa. Since it typically takes say 1% of the Young's modulus (number vary quite a bit and are scale dependent, but to give a rough guess) in Cauchy stress to induce plasticity, the reported numbers make perfect sense, but the explanations are not to the point (as in other cases)."

Reviewer #3 is missing the point. Their manuscript is not an attempt to explain the low friction of graphite in general. Rather, they are showing that the low friction of SSL contacts with graphite can persist for very long times under high pressure. The authors could address this with some additional comments regarding the mechanism above. Note that graphite as a macroscale solid lubricant fails in dry or vacuum conditions; some water is needed because defects are inevitably formed, and interfacial bonds form across the interface, leading to wear. The presence of water is believed to provide -H and -OH terminations when C-C bonds break, preventing interfacial bonds from forming. The fact that such interfacial bonds can be prevented over long sliding periods is what is new here, and distinct from what Reviewer #3 was concerned about.

In addition to the above changes needed (simulations of WO_x , more information about the tip's roughness and surface composition), the authors should address these additional concerns:

Major concerns:

4. The authors should comment on the expected density of grains on the mesa and substrate. What is the grain size of the HOPG used to form the mesa, and the "normal-flake-graphite (NGS, Germany)" used to form the substrate? This is central to their claim of SSL - that they have two perfect crystalline regions with no grain boundaries. How do they know this? Have they measured the presence of higher friction when a grain boundary is present (or, perhaps more interestingly, is the friction still low if a GB is present)?

5. Related somewhat to the above point, the authors launch into the discussion of wear resistance in the experimental results before demonstrating that the substrate is superlubric. They should show and discuss Fig. 1f before showing and discussing the wear results in Fig. 1d. They should provide a number for the friction coefficient and the shear strength (I calculated the shear strength by hand and came up with ~ 25 kPa, which is indeed superlubric). This is particularly important because, while their friction coefficient is low, the friction vs. pressure plot does not go through the origin (presumably because of adhesion).

6. Related to this, the statement on page 5, "Notably, our recent efforts ... defined SSL at the large-scale as the contact formed by atomically smooth interfaces experiencing vdW interactions with a friction coefficient on the order of 10^{-3} (ref 32)", is problematic. Since there is adhesion, a low friction coefficient is not enough; the shear strength should be low as well. Admittedly this requires an ad-hoc choice for a "low" value of shear strength, but atomically-smooth interfaces have been reported with no pressure-dependence to the shear strength (i.e. nearly zero "friction coefficient") but with shear strengths reaching the ideal limit! The authors should clarify this point.

7. More information needs to be provided regarding the cleaning protocol. A height scale must be given for the AFM images in Fig. 2d, and the difference in mean height (\pm the standard deviation) between sliding region and the outside region should be provided. Please provide the phase image that goes along with Fig. 2d. Does this provide any clues on what the debris is?

8. The authors present DFT results for a large number of materials that are not tested in the experiments, observing a correlation between interfacial bond formation and both the bulk modulus and the first ionization potential. This really reads like a separate piece of work. The alternate materials are not relevant to understand the experimental results (in fact, the authors don't even simulate the actual material used in the experiment - tungsten oxide). The results are interesting but I don't see how it helps explain the experimental results, and none of the DFT

results are validated with experiments on any of these alternate materials. This section should be removed. This will streamline the paper.

Minor concerns:

9. I recommend the authors avoid the term "symmetric" and "asymmetric" contact. Those terms are used in the literature, but since SSL involves symmetry, it can cause confusion. "Self-mated" and "non self-mated" is better.

10. Page 2: the low friction seen for MoS₂ (ref 13) may not be due to SSL. Even when commensurate, MoS₂-MoS₂ sliding friction is low. This is "superlubricity", i.e. a friction coefficient less than 0.01, but cannot be claimed as SSL.

11. Page 2: Archard's model is not a gradual wear model (gradual implies low; Archard holds for high wear cases as well. Perhaps the authors mean "progressive"?).

12. The authors say the "asymmetric SSL contact is expected to be less pressure-resistant (ref 45)" It's not clear why this follows from Ref. 45; a brief explanatory statement is needed. Or, the sentence can be removed.

13. It's interesting that friction does not increase substantially (and maybe even decreases, according to Fig. 1f) once the threshold pressure for wear is reached. Can the authors comment on why this is? It's a nice result; low friction persists even when damage starts. That said, should higher friction be expected if interfacial bonding and damage are occurring?

14. The authors mention on page 11 that graphite is hydrophobic. That is only true when it is contaminated by hydrocarbon adsorbates which are ubiquitous in ambient conditions. However, clean graphite is somewhat hydrophilic (see Li, Z., Wang, Y., Kozbial, A., Shenoy, G., Zhou, F., McGinley, R., Ireland, P., Morganstein, B., Kunkel, A., Surwade, S.P. and Li, L., 2013. Effect of airborne contaminants on the wettability of supported graphene and graphite. *Nature materials*, 12(10), pp.925-931.)

RESPONSE TO REVIEWERS' COMMENTS

Reviewer #1

The authors have responded well to the reviewer's comments, and I recommend that the manuscript be accepted for publication.

Reply: Thank you for the recommendation.

Reviewer #2

The authors have addressed the points raised by the referees and the manuscript is improved. I recommend to publish this paper in its present form.

Reply: Thank you for the recommendation.

Reviewer #4

The authors addressed my comments/questions in quite thorough matter. I also see that the critique from the other reviewers is well considered and is addressed both in the authors' responses and in revisions to the manuscript. This includes also their reasonable argument regarding the degree of novelty raised by the R3. The work establishes a nonlinear critical change in lubricity to friction and wear beyond certain critical pressure, a finding of significance to the field and related fields, perhaps for many other material-pairs and interfaces. Overall, I find the manuscript now in good shape and worthy acceptance for publication in Nt. Comm. in present form.

Reply: Thank you for the recommendation.

Reviewer #3

While I appreciate the experiments presented by Sun, Gao, Jia, and coworkers, and see some of my comments addressed, I still do not believe that this paper is a significant step forward in our understanding of superlubricity and/or the onset of wear in graphite or hard materials. Even worse, the more I think about the claimed novelties of the paper, the less I recognize them and the more problems I see regarding the modeling of the paper.

Reply: Thank you for the follow-up concerns on our manuscript, which was addressed in our revision. The novelty of our work is the first experimental demonstration of wear-free SSL over large sliding distances at high pressures at ambient conditions. We hope our revised manuscript offers a better presentation in clarifying the points made in your comments.

First, correlating ionization energies with stiffness of solids goes as far back as to Slater, though I apologize for not remembering the precise reference. Correlations of bulk and shear modulus are commonly reported in works on interatomic potentials, etc.

Reply: Thank you for pointing out the classical works on the correlation between chemical and mechanical properties of solids, which were aware by us. Our discussion extends these understandings to the interface between graphite and metals, by identifying the critical pressures where transitions in interfacial interaction are triggered, which has not been reported to the best of our knowledge.

Changes made:

Comments were added to the main text on the correlation between ionization energies and the stiffness of solids.

‘It is well known that the ‘physical’ stiffness of a solid is strongly tied to the ‘chemical’ one defined by the ionization energy (IE) and the electron affinity (EA) [58]. This understanding is extended to the vdW interface here.’

Second, I am not sure that much can be learned from a theoretical analysis of perfectly flat tungsten without oxide layer squeezed against graphite. Both unavoidable roughness and termination with an oxide layer have an extreme effect on the interaction.

Reply: Thank you for mentioning the effects of the oxide layers, which was discussed in this revision by additional experimental and theoretical work. First, as our finite element analysis (FEA) shows, the central contact with high local pressure is rather limited (hundreds of nanometers), which can be considered to be locally flat. This argument is verified by the fact that the critical pressure obtained from FEA is consistent with the value estimated from density functional theory (DFT) calculations. Previous work showed that the W tip etched by KOH is smooth (root mean square (RMS) roughness < 0.3 nm, Fig. R1(a) [1]). We also characterized our

tip using high-resolution SEM and AFM. The results show height variation of a few nanometers at the micrometer scale and atomistic smoothness at the nanometer scale (Fig. R1(d)), which validate the argument that local roughness is not a crucial issue. A recent work [2] shows that the change in contact pressure converges as the wavelength of roughness increases. Therefore, the effect of roughness on contact pressure is negligible in our experiments, and our first-principles calculations using a flat contact can thus be used for the discussion. We added this discussion on the roughness to the revision.

Fig. R1 | High-resolution scanning electron microscopy (SEM) images of KOH-etched W tips reported in Ref. [1] (a) and in our experiments (b). (c), AFM image of the tip. (d), A line scan profile taken from the line labeled in (c), which shows height variation of a few nanometers at the micrometer scale and atomistic smoothness at the nanometer scale.

Secondly, the oxide layer can have a strong impact on the nature of contact. In order to assess the quantitative effect, we firstly characterized the oxide, and following structural characterization in the literature, we carried out DFT calculations and find that the critical pressure changes from 5.4 GPa for the W/graphite interface to 3.5 GPa for the WO₃/graphite interface. Both results are included in the manuscript to understand the effect of oxide layers. Thank you for the insightful comments that allow us to improve the work by adding the discussion of oxide effects.

Fig. R2 | X-ray photoelectron spectroscopy (XPS) characterization of the tungsten tip etched by KOH. In our experiments, the tungsten tip is used days and weeks after the etching process. The XPS characterization was thus conducted 3 days after etching.

Changes made:

The subsection ‘Mechanisms of the breakdown process and wear of SSL’ was updated by using the results of WO₃/graphite for discussion.

‘Wear at the SSL contact can be directly related to the pressure enforced across the interface, the mechanochemistry of which was explored by first-principles calculations based on the density functional theory (DFT). X-ray photoelectron spectroscopy (XPS) characterization of the tungsten tip suggests the existence of an oxide layer over the tip surface (Fig. S6(f)). A model consisting of a ($\sqrt{2}\times\sqrt{2}$)R45°-reconstructed (001) WO₃ surface and a graphite substrate, which closely follows our experimental setup and previous studies [49, 50] (Fig. 3(a, b)). The DFT calculation results show that the pressure increases rapidly with the compression and declines after the peak at a pressure of 3.50 GPa (Fig. 3(d)), which is defined as the breakdown pressure (P_{cr}) due to the formation of interfacial O-C bonds (Fig. 3(c)). The value of estimated critical pressure agrees with the experimental value of 3.74 GPa.’

The discussion on the roughness was added to the results and discussion.

‘Previous work [47] showed that the tungsten tip etched by KOH is smooth, with a root mean square (RMS) roughness below 0.3 nm (Fig. S6). We characterized our tip using high-resolution SEM and AFM. The results show height variation of a few nanometers at the micrometer scale and atomistic smoothness at the nanometer scale (Fig. S6(c, d)), which validates the argument that local roughness is not a crucial issue for the discussion. A recent work [48] shows that the contact pressure change converges as the roughness wavelength increases. Therefore, the effect of roughness on contact pressure is negligible.’

Third, the authors fail to deliver the simplicity of the real argument and the reason why graphite has been used as lubricant for more than a century. Its softness in the normal direction makes graphite have relatively small contact pressures (which the authors overcome by using hard tips and localized forces, the strong part of the work) given a rough counterface. Within the plane, bonds are stiff due to the sp² hybridization yielding in-plane moduli in the order of TPa. Since it typically takes say 1% of the Young's modulus (number vary quite a bit and are scale dependent, but to give a rough guess) in Cauchy stress to induce plasticity, the reported numbers make perfect sense, but the explanations are not to the point (as in other cases).

Reply: Thank you for offering the suggestions on the general discussion on graphite in lubricant applications. The anisotropy of graphite and softness in the *c*-axis direction can increase the contact area under concentrated loads as explained in your comments. The indentation approach is used in our work to induce a localized force. We added this discussion to the main text by following your suggestions.

Changes made:

Discussion on pressure localization was added to the main text.

'Our indentation setup in the sliding tests results in a local high pressure across an effective contact region of hundreds of nanometers (Supplementary Note 1).'

Reviewer #5

I have reviewed the manuscript by Sun et al, along with the supplementary material, the referee reports, the rebuttal, and the subsequent referee correspondence. My view is that this manuscript can be suitable for publication in Nat. Comm. if certain issues can be addressed, which are listed further below. First, regarding overall novelty, the experimental work is novel. SSL has not been shown to be wear-free over large sliding distances at high pressures in ambient conditions. This is a significant advance and is the key point of novelty of the work. The authors state in response to Reviewer 3 that "To our knowledge, there is no prior paper that actually reported such wear behavior and mechanism of SSL contacts". I agree with this statement.

Reply: Thank you for highlighting the novelty of our work and kindly offering insightful comments that helped us to improve the work. Please find below our point-to-point replies to the comments.

Regarding Reviewer #3's remaining concerns, I have the following comments:

1. *"First, correlating ionization energies with stiffness of solids goes as far back as to Slater, though I apologize for not remembering the precise reference. Correlations of bulk and shear modulus are commonly reported in works on interatomic potentials, etc."*

The discussion of this point in the manuscript is in need of revision as I specify below, but is not a reason to reject the paper.

Reply: Thank you for pointing out the work on the correlation between chemical and mechanical properties of solids, which were aware by us. Our discussion extends these understandings to the interface between graphite and metals, and the critical pressures where the transition in interfacial interaction are triggered, which has not been reported to the best of our knowledge.

Changes made:

Comments were added to the main text on the correlation between ionization energies and the stiffness of solids.

'It is well known that the 'physical' stiffness of a solid is strongly tied to the 'chemical' one defined by the ionization energy (IE) and the electron affinity (EA) [55]. This understanding is extended to the vdW interface here.'

2. *"Second, I am not sure that much can be learned from a theoretical analysis of perfectly flat tungsten without oxide layer squeezed against graphite. Both unavoidable roughness and termination with an oxide layer have an extreme effect on the interaction."*

2.1 The referee has valid points here, but I expect they are addressable. The DFT presented in the paper for clean W is not relevant for the results. They should simulate the contact of tungsten oxide with graphite, not metallic W. The native oxide of W is reported in literature to be ~5 nm

thick. I'm not sure of its stoichiometry or structure, but the authors should check the literature or, even better, do surface science on their W (doing XPS on the W material etched with the same KOH procedure should suffice; it's hard to do it on the tip itself although TEM could achieve this) to determine the oxidation state of the W. Then they should simulate this. I suspect they will see that covalent bonds still can form, but probably under even higher pressures since the W atoms will be less likely to bond with C when in higher oxidation states. Defects in the graphite, as the authors show, would lead to lower pressures where wear occurs, in line with experiments. In short, the simulations should match the experiments. The end of the tip is not W, it's WO_x . The O adsorbates simulated in Fig. S13 are not sufficient; they should simulate an oxide, not just a few O adsorbates.

Reply: Thank you for the insightful comment. By following the instruction you suggested, we used X-ray photoelectron spectroscopy (XPS) to characterize the surface of the KOH-etched W tip used in our experiments. The results clearly indicate the formation of oxides in the form of WO_3 (Fig. R2), which is consistent with the previous experimental evidence [3]. We then conducted first-principles calculations for the contact with a $(\sqrt{2} \times \sqrt{2})R45^\circ$ -reconstructed (001) surface of WO_3 , which features the lowest surface energy [4-5]. The discussion on the W surface is still included in the discussion for comparison. The critical pressure changes from 5.4 GPa for the W/graphite interface to 3.5 GPa for the WO_3 /graphite interface, and our conclusion on the pressure-induced transition in the interlayer bonding features remains valid.

Fig. R2 | X-ray photoelectron spectroscopy (XPS) characterization of the tungsten tip etched by KOH. In our experiments, the tungsten tip is used days and weeks after the etching process. The XPS characterization was thus conducted 3 days after etching.

Changes made:

The subsection 'Mechanisms of the breakdown process and wear activation' was updated by using the results of WO_3 /graphite for discussion.

‘Wear at the SSL contact can be directly related to the pressure enforced across the interface, the mechanochemistry of which was explored by first-principles calculations based on the density functional theory (DFT). X-ray photoelectron spectroscopy (XPS) characterization of the

tungsten tip suggests the existence of an oxide layer over the tip surface (Fig. S6(f)). A model consisting of a $(\sqrt{2}\times\sqrt{2})R45^\circ$ -reconstructed (001) WO_3 surface and a graphite substrate, which closely follows our experimental setup and previous studies [49, 50] (Fig. 3(a, b)). The DFT calculation results show that the pressure increases rapidly with the compression and declines after the peak at a pressure of 3.50 GPa (Fig. 3(d)), which is defined as the breakdown pressure (P_{cr}) due to the formation of interfacial O-C bonds (Fig. 3(c)). The value of estimated critical pressure agrees with the experimental value of 3.74 GPa.'

2.2 Regarding roughness, this is a universal problem in experiments: even a "smooth" tip will have atomic-scale roughness from steps, defects, grain boundaries, or amorphous regions that will complicate the contact mechanics. However, the graphite is rather compressible in the vertical direction and this may well lead to relatively conformal contact. This is not a reason to reject the paper. If there is literature on the roughness of W tips etched in the same manner as the present manuscript, the authors could refer to that for support (i.e., perhaps others have done high-res SEM or TEM of such etched tips in the past). Alternately the authors could try to take an AFM image of the end of the tip (not easy, but actually done by many groups previously). The optical is not sufficient (Fig. S8) for this; it's a start, as it rules out gross transfer of C to the tip, but more is needed to confirm the tip is reasonably smooth. It could be valuable to take the overall W tip radius and the roughness power spectrum for the W, and use the model of Robbins & Pastewka for rough round tips with self-affine surfaces to get an estimate of the contact pressures. This is not required, but would strengthen the paper.

Reply: Thank you for the suggestions. Previous work showed that the W tip etched by KOH is smooth (root mean square (RMS) roughness < 0.3 nm, Fig. R1(a) [1]). We also characterized our tip using high-resolution SEM and AFM. The results show height variation of a few nanometers at the micrometer scale and atomistic smoothness at the nanometer scale (Fig. R1(d)), which validates the argument that local roughness is not a crucial issue. A recent work [2] shows that the change in contact pressure converges as the wavelength of roughness increases. Therefore, the effect of roughness on contact pressure is negligible in our experiments, and our first-principles calculations using a flat contact can thus be used for the discussion.

Fig. R1 | High-resolution scanning electron microscopy (SEM) images of KOH-etched W tips reported in Ref. [1] (a) and in our experiments (b). c, AFM image of the tip. d, A line scan profile taken from the line labeled in (c), which shows height variation of a few nanometers at the micrometer scale and atomistic smoothness at the nanometer scale.

Changes made:

The discussion on the roughness was added to the results and discussion.

‘Previous work [47] showed that the tungsten tip etched by KOH is smooth, with a root mean square (RMS) roughness below 0.3 nm (Fig. S6). We characterized our tip using high-resolution SEM and AFM. The results show height variation of a few nanometers at the micrometer scale and atomistic smoothness at the nanometer scale (Fig. S6(c, d)), which validates the argument that local roughness is not a crucial issue for the discussion. A recent work [48] shows that the contact pressure change converges as the roughness wavelength increases. Therefore, the effect of roughness on contact pressure is negligible.’

2.3 Regarding contaminants: again, this is a universal problem in experiments and more importantly, such contaminants will be present in typical applications. The cleaning procedure the authors show is sufficient to address this. I recommend they include Fig. R6 from the response to Reviewer 4 in the Supplementary Material to further support their argument about

the effectiveness of their cleaning procedure. That said, they need to modify their language about this issue. They surely remove some molecules, but they do not know how much (no one does! It's extremely difficult to measure this). The statement on page 12 that "the pre-cleaning step before tests can remove the confined molecules" should be modified to something like "the pre-cleaning step before tests removes some of the confined molecules, reducing friction to a steady-state level" (and refer to Fig. R6). An additional comment acknowledging that contaminants may still be present is needed, since they cannot rule that out. Finally, the statement "The influence of the intercalant molecules is studied by simulations (Supplementary Notes 6-7)..." needs to be removed; Supplementary Notes 6 and 7 say nothing about contaminant molecules. They only simulate vacancy defects and oxygen adsorbates. The contaminants are likely hydrocarbons and water.

Reply: Thank you for the suggestions to improve the presentation. We included **Fig. R6** in our previous reply to comments to the revised Supplementary Material (**Fig. S3**) to clarify the details of cleaning procedures. We also added more discussion on the contaminants by following your suggestion.

Changes made:

Fig. S3 was revised.

Fig. S3 | Effect of the pre-cleaning step (a-c) and phase image of debris (d, e). (a) AFM image of the graphite substrate after a pre-cleaning procedure. The white box annotates the pre-cleaned region. The yellow box annotates the phase image shown in the right side figure. (b) Friction evolution with increasing sliding cycles. (c) Surface roughness comparison of the graphite substrate at the cleaned region and the outside region annotated by solid lines in (a). (d) AFM

phase image of the graphite substrate after sliding tests of the tungsten/graphite contact. (e) Magnified phase image of the debris indicated by the dashed box in (d).

Discussion on the contaminants was added.

‘A pre-cleaning step was carried out to sweep out contaminants (e.g., adsorbed molecules such as water and hydrocarbons [39–41], see Supplementary Note 2 for details). The contaminants originate from the environment before the construction of the contact and may not be completely excluded at the contact. However, the ultra-low friction coefficient of SSL is still preserved, indicating the robustness of SSL against the atmosphere [41]. Consequently, the pre-cleaning step before tests removes some of the confined molecules, reducing friction to a steady-state level (Fig. S3). Our previous work reported no obvious difference in friction of SSL in the mesa/substrate setup in ambient conditions and nitrogen atmosphere with relative humidities (RHs) of 42% and 10%, respectively [19].’

3. *"Third, the authors fail to deliver the simplicity of the real argument and the reason why graphite has been used as lubricant for more than a century. Its softness in the normal direction makes graphite have relatively small contact pressures (which the authors overcome by using hard tips and localized forces, the strong part of the work) given a rough counterface. Within the plane, bonds are stiff due to the sp^2 hybridization yielding in-plane moduli in the order of TPa. Since it typically takes say 1% of the Young's modulus (number vary quite a bit and are scale dependent, but to give a rough guess) in Cauchy stress to induce plasticity, the reported numbers make perfect sense, but the explanations are not to the point (as in other cases)."*

Reviewer #3 is missing the point. Their manuscript is not an attempt to explain the low friction of graphite in general. Rather, they are showing that the low friction of SSL contacts with graphite can persist for very long times under high pressure. The authors could address this with some additional comments regarding the mechanism above. Note that graphite as a macroscale solid lubricant fails in dry or vacuum conditions; some water is needed because defects are inevitably formed, and interfacial bonds form across the interface, leading to wear. The presence of water is believed to provide -H and -OH terminations when C-C bonds break, preventing interfacial bonds from forming. The fact that such interfacial bonds can be prevented over long sliding periods is what is new here, and distinct from what Reviewer #3 was concerned about.

Reply: Thank you for clarifying this point. We added relevant discussions on the lubricant behaviors of graphite to the introduction for the completeness of presentation. We also highlight our main point on the wearless performance of graphite SSL contacts under GPa-level pressures.

Changes made:

Discussion on the lubrication mechanism of graphite was added.

‘This finding agrees with the fact that solid lubrication using graphite should avoid the formation of interlayer bonding in dry or vacuum conditions, where water can solve this problem by providing -H and -OH terminations when C-C bonds break.’

In addition to the above changes needed (simulations of WO_x , more information about the tip's roughness and surface composition), the authors should address these additional concerns:

Reply: Thank you for summarizing the revisions we need to improve the work. In addition to addressing the issues of oxide layers, roughness, and contaminants and further clarifying the major point of our presentation, we addressed your following constructive comments by carrying out more experimental work. Please find below the details.

Major concerns:

4. The authors should comment on the expected density of grains on the mesa and substrate. What is the grain size of the HOPG used to form the mesa, and the "normal-flake-graphite" (NFG, from NGS, Germany) used to form the substrate? This is central to their claim of SSL - that they have two perfect crystalline regions with no grain boundaries. How do they know this? Have they measured the presence of higher friction when a grain boundary is present (or, perhaps more interestingly, is the friction still low if a GB is present)?

Reply: Thank you for the comments and suggestions. We used electron back scatter diffraction (EBSD) to characterize the grain texture of graphite (Fig. R3), which confirms that the surfaces in contact are single-crystalline and free of grain boundaries. These experimental results and relevant discussion were added to the revision.

Fig. R3 | EBSD results of graphite. (a-c) Band contrast (BC) and inverse pole figures (IPF) of HOPG. (d,e) BC and IPF images of the normal-flake graphite.

Changes made:

Fig. R3 was added to Fig. 1.

Discussion on the texture of graphite was added.

‘Our previous works showed that any cleaved SRM flake from highly oriented pyrolytic graphite (HOPG) features single-crystalline surfaces without detectable defects, as characterized by atomic force microscopy (AFM), electron backscattered diffraction (EBSD), and Raman spectrum techniques [19, 36, 37]. Crystal orientations and surface roughness of the graphite substrate were studied using EBSD (Fig. 1 (f-m), OM, and AFM (Fig. S2) before tests. These results confirm that the surfaces in contact are single-crystalline and free of grain boundaries.’

5. Related somewhat to the above point, the authors launch into the discussion of wear resistance in the experimental results before demonstrating that the substrate is superlubric. They should show and discuss Fig. 1f before showing and discussing the wear results in Fig. 1d. They should provide a number for the friction coefficient and the shear strength (I calculated the shear strength by hand and came up with ~25 kPa, which is indeed superlubric). This is particularly important because, while they friction coefficient is low, the friction vs. pressure plot does not go through the origin (presumably because of adhesion).

Reply: Thank you for the suggestion on writing. We rearranged the introduction of experimental results in the main text and added relevant discussion on the friction coefficients. The friction force in **Fig. 1c** was converted to the shear strength, which is ~20 kPa.

Changes made:

The friction stress and shear strength were added to the discussion.

‘The friction stress between the mesa and the substrate is ultralow (~20 kPa), and the friction coefficient (the ratio of the friction force and the normal force is on the order of 10^{-5} under pressures between $P = 1$ GPa to 6 GPa (Fig. 1(c), see Supplementary Note 1 for details). The nearly pressure-independence of friction forces measured up to the gigapascal scale demonstrates the robustness of SSL at the graphite/graphite contact under high pressure.’

6. Related to this, the statement on page 5, "Notably, our recent efforts ... defined SSL at the large-scale as the contact formed by atomically smooth interfaces experiencing vdW interactions with a friction coefficient on the order of 10^{-3} (ref 32)", is problematic. Since there is adhesion, a low friction coefficient is not enough; the shear strength should be low as well. Admittedly this requires an ad-hoc choice for a "low" value of shear strength, but atomically-smooth interfaces have been reported with no pressure-dependence to the shear strength (i.e. nearly zero "friction coefficient") but with shear strengths reaching the ideal limit! The authors should clarify this point.

Reply: Thank you for raising this important point. In defining SSL, a low shear strength is required if the mechanical energy dissipation is related directly to it, instead of the friction

coefficient by considering the situation where no pressure is added. We added our measurement on the shear strength and clarify the definition of SSL by adding these details.

Changes made:

Discussion on the shear strength was added,

‘The friction coefficient is measured to be on the order of 10^{-3} even under gigapascal-level pressures (Fig. S5). However, the shear strength, $\tau_s = 10$ MPa, is much higher than that of the graphite contact (~ 20 kPa) possibly due to the adhesion effect. Consequently, the notion of SSL should be justified if the mechanical energy dissipation during the sliding process and thus the shear strength are of concern.’

7. More information needs to be provided regarding the cleaning protocol. A height scale must be given for the AFM images in Fig. 2d, and the difference in mean height (\pm the standard deviation) between sliding region and the outside region should be provided. Please provide the phase image that goes along with Fig. 2d. Does this provide any clues on what the debris is?

Reply: Thank you for the comments. The height scale of AFM images was added. We also provided more information regarding the cleaning protocol in the supplementary materials. The results demonstrate the effective reduction of the confined intercalant molecules at contact and clues of the derivation of debris. The phase images were added to the revision, from which we cannot tell the chemistry of the debris, unfortunately. However, from previous studies, they are expected to be airborne hydrocarbons and water molecules [6-10].

Changes made:

The height scales in Figs. 1 and 2 were added.

The phase image was added to Fig. S3.

Discussion on the contaminants was revised.

‘A pre-cleaning step was carried out to sweep out contaminants (e.g., adsorbed molecules such as water and hydrocarbons [39–41], see Supplementary Note 2 for details). The contaminants originate from the environment before the construction of the contact and may not be completely excluded at the contact. However, the ultra-low friction coefficient of SSL is still preserved, indicating the robustness of SSL against the atmosphere [41]. Consequently, the pre-cleaning step before tests removes some of the confined molecules, reducing friction to a steady-state level (Fig. S3). Our previous work reported no obvious difference in friction of SSL in the mesa/substrate setup in ambient conditions and nitrogen atmosphere with relative humidities (RHs) of 42% and 10%, respectively [19].’

8. The authors present DFT results for a large number of materials that are not tested in the experiments, observing a correlation between interfacial bond formation and both the bulk

modulus and the first ionization potential. This really reads like a separate piece of work. The alternate materials are not relevant to understand the experimental results (in fact, the authors don't even simulate the actual material used in the experiment - tungsten oxide). The results are interesting but I don't see how it helps explain the experimental results, and none of the DFT results are validated with experiments on any of these alternate materials. This section should be removed. This will streamline the paper.

Reply: Thank you for the suggestion. We revised Subsection ‘Understanding SSL robustness under high pressures’ to limit the discussion on the theoretical insights. The goal of the discussion is to understand the unexpected robustness of SSL states and extend our discussion based on experimental results to a wide spectrum of materials that may be considered in SSL-enable applications. The discussion on bare metal surfaces in absence of oxidation complement the discussion on the WO₃/graphite contact.

Changes made:

Subsection ‘Understanding SSL robustness under high pressures’ was updated.

‘Our work demonstrates a novel wear-free feature of the graphitic SSL contacts without interfacial bonds under GPa pressures over long sliding periods. To understand the unexpected robustness of SSL states and extend our discussion to SSL-enable applications, we studied the material dependence. A W/graphite contact can be constructed by preventing oxidation and was studied by performing DFT calculations for comparison with the WO₃/graphite and graphite/graphite contacts (see Methods for details). The DFT calculation results suggest a higher breakdown pressure of $P_{cr} = 5.4$ GPa for the contact with bare W (001) surface. We find that instead of covalent bonding at the WO₃/graphite contact beyond P_{cr} , the transition in the electronic coupling at the W/graphite interface is mediated by charge transfer [54, 55], and this electrostatic nature of interaction results in a higher value of P_{cr} .’

Minor concerns:

9. I recommend the authors avoid the term "symmetric" and "asymmetric" contact. Those terms are used in the literature, but since SSL involves symmetry, it can cause confusion. "Self-mated" and "non self-mated" is better.

Reply: Thank you for the suggestion. We replaced the terms "symmetric" and "asymmetric" contact to "self-mated" and "non-self-mated" contact.

10. Page 2: the low friction seen for MoS₂ (ref 13) may not be due to SSL. Even when commensurate, MoS₂-MoS₂ sliding friction is low. This is "superlubricity", i.e. a friction coefficient less than 0.01, but cannot be claimed as SSL.

Reply: Thank you for the comment. We removed Ref. 13.

11. Page 2: Archard's model is not a gradual wear model (gradual implies low; Archard holds for high wear cases as well. Perhaps the authors mean "progressive"?).

Reply: Thank you for the comment. We replaced the term "gradual wear" to "progressive wear" in the main text.

12. The authors say the "asymmetric SSL contact is expected to be less pressure-resistant (ref 45)" It's not clear why this follows from Ref. 45; a brief explanatory statement is needed. Or, the sentence can be removed.

Reply: Thank you for the suggestion. Reference 45 analyses the copper (111)-graphene and nickel (111)-graphene contacts, which are characterized as non-self-mated contacts, indicating a low breakdown pressure (5 and 18 GPa, respectively). These values are much smaller than the graphite-graphite contact (160 GPa). We clarified this point in the revision.

Changes made:

The pressure resistance of asymmetric SSL contacts was discussed.

'Theoretical calculations demonstrate that non-self-mated SSL contacts between metal and graphene exhibit weaker pressure resistance than the graphite-graphite contacts [44], and it is natural to question the robustness of the SSL state therein.'

13. It's interesting that friction does not increase substantially (and maybe even decreases, according to Fig. 1f) once the threshold pressure for wear is reached. Can the authors comment on why this is? It's a nice result; low friction persists even when damage starts. That said, should higher friction be expected if interfacial bonding and damage are occurring?

Reply: Thank you for your interest. Actually, **Fig. 1f** summarizes the friction at the graphite/graphite interface under a pressure below 6 GPa, which is well below the critical pressure (> 9.45 GPa). We clarified this point in the revision.

Changes made:

Figure Caption 1 was revised.

‘(c) Experimental measurement of the average shear stress below the breakdown pressure.’

14. The authors mention on page 11 that graphite is hydrophobic. That is only true when it is contaminated by hydrocarbon adsorbates which are ubiquitous in ambient conditions. However, clean graphite is somewhat hydrophilic (see Li, Z., Wang, Y., Kozbial, A., Shenoy, G., Zhou, F., McGinley, R., Ireland, P., Morganstein, B., Kunkel, A., Surwade, S.P. and Li, L., 2013. Effect of airborne contaminants on the wettability of supported graphene and graphite. *Nature materials*, 12(10), pp.925-931.)

Reply: Thank you for the comment. We revised our discussion on the hydrophobicity of graphite.

Changes made:

‘A pre-cleaning step was carried out to sweep out contaminants (e.g., adsorbed molecules such as water and hydrocarbons [39-41], see Supplementary Note 2 for details). The contaminants originate from the environment before the construction of the contact and may not be completely excluded at the contact. However, the ultra-low friction coefficient of SSL is still preserved, indicating the robustness of SSL against the atmosphere [41]. Consequently, the pre-cleaning step before tests removes some of the confined molecules, reducing friction to a steady-state level (Fig. S3). Our previous work reported no obvious difference in friction of SSL in the mesa/substrate setup in ambient conditions and nitrogen atmosphere with relative humidities (RHs) of 42% and 10%, respectively [19].’

References

- [1] D. Xu, K. M. Liechti, K. Ravi-Chandar, Mesoscale scanning probe tips with subnanometer RMS roughness. *Rev. Sci. Instrum.* 78(7) (2007)
- [2] J. M. Monti, L. Pastewka, M. O. Robbins, Fractal geometry of contacting patches in rough elastic contacts. *J. Mech. Phys. Solids* 160, 104797 (2022)
- [3] A. Dunand, M. Minissale, J.-B. Faure, et al., Surface oxygen versus native oxide on tungsten, Contrasting effects on deuterium retention and release. *Nucl. Fusion* 62(5), 054002 (2022)
- [4] P. M. Oliver, S. C. Parker, R. G. Egdell, F. H. Jones, Computer simulation of the surface structures of WO_3 . *J. Chem. Soc., Faraday Trans.* 92(12), 2049-2056 (1996)
- [5] J. Meng, Z. Lan, I. E. Castelli, K. Zheng, Atomic-scale observation of oxygen vacancy-induced step reconstruction in WO_3 . *J. Phys. Chem. C* 125(15), 8456-8460 (2021)
- [6] P. Gallagher, Y. Li, K. Watanabe, T. Taniguchi, T. F. Heinz, D. Goldhaber-Gordon, Optical imaging and spectroscopic characterization of self-assembled environmental adsorbates on graphene. *Nano Lett.* 18(4), 2603-2608 (2018)
- [7] A. Pálincás, G. Kálvin, P. Vancsó, et al., The composition and structure of the ubiquitous hydrocarbon contamination on van der Waals materials. *Nat. Commun.* 13(1), 6770 (2022)
- [8] R. Bai, N. L. Tolman, Z. Peng, H. Liu, Influence of atmospheric contaminants on the work function of graphite. *Langmuir* 39(34), 12159-12165 (2023)
- [9] C. A. Amadei, C.-Y. Lai, D. Heskes, M. Chiesa, Time dependent wettability of graphite upon ambient exposure: The role of water adsorption. *J. Chem. Phys.* 141(8), 084709 (2014)
- [10] J.-Y. Lu, C.-Y. Lai, I. Almansoori, M. Chiesa, The evolution in graphitic surface wettability with first-principles quantum simulations: The counterintuitive role of water. *Phys. Chem. Chem. Phys.* 20(35), 22636-22644 (2018)

REVIEWERS' COMMENTS

Reviewer #5 (Remarks to the Author):

I am satisfied with all of the authors' responses and I support publishing this manuscript in Nat. Comm.